# An engineered transcriptional reporter of protein localization identifies regulators of mitochondrial and ER membrane protein trafficking in high-throughput CRISPRi screens

**Robert Coukos[1†], David Yao[1†], Mateo I Sanchez[1,2], Eric T Strand[1], Meagan E Olive[3], Namrata D Udeshi[3], Jonathan S Weissman[4,5,6,7], Steven A Carr[3], Michael C Bassik[1]\*, Alice Y Ting[1,2,8]\***

[1]Department of Genetics, Stanford University, Stanford, United States; [2]Chan Zuckerberg Biohub, Stanford, United States; [3]Broad Institute of MIT and Harvard, Cambridge, United States; [4]Whitehead Institute, Cambridge, United States; [5]Department of Biology, Massachusetts Institute of Technology, Cambridge, United States; [6]Department of Cellular and Molecular Pharmacology, University of California, San Francisco, San Francisco, United States; [7]Howard Hughes Medical Institute, University of California, San Francisco, San Francisco, United States; [8]Department of Biology, Stanford University, Stanford, United States

**\*For correspondence:**
bassik@stanford.edu (MCB);
ayting@stanford.edu (AYT)

[†]These authors contributed equally to this work

**Competing interest:** The authors declare that no competing interests exist.

**Abstract** The trafficking of specific protein cohorts to correct subcellular locations at correct times is essential for every signaling and regulatory process in biology. Gene perturbation screens could provide a powerful approach to probe the molecular mechanisms of protein trafficking, but only if protein localization or mislocalization can be tied to a simple and robust phenotype for cell selection, such as cell proliferation or fluorescence-activated cell sorting (FACS). To empower the study of protein trafficking processes with gene perturbation, we developed a genetically encoded molecular tool named HiLITR (High-throughput Localization Indicator with Transcriptional Readout). HiLITR converts protein colocalization into proteolytic release of a membrane-anchored transcription factor, which drives the expression of a chosen reporter gene. Using HiLITR in combination with FACS-based CRISPRi screening in human cell lines, we identified genes that influence the trafficking of mitochondrial and ER tail-anchored proteins. We show that loss of the SUMO E1 component SAE1 results in mislocalization and destabilization of many mitochondrial tail-anchored proteins. We also demonstrate a distinct regulatory role for EMC10 in the ER membrane complex, opposing the transmembrane-domain insertion activity of the complex. Through transcriptional integration of complex cellular functions, HiLITR expands the scope of biological processes that can be studied by genetic perturbation screening technologies.

## Introduction

Gene perturbation screens, in which libraries of cells bearing individual genetic perturbations are assessed for fitness, growth, or other phenotypes, have been broadly applied to uncover the genetic bases of specific cellular processes. For example, CRISPR-based knockout screens have identified essential human genes (*Hart et al., 2015*; *Shalem et al., 2014*; *Wang et al., 2015*), discovered factors that confer drug or toxin resistance (*Wang et al., 2014*; *Zhou et al., 2014*), and dissected signaling

and regulatory networks (*Klann et al., 2017*; *Parnas et al., 2015*). In general, gene perturbation screens can be implemented in either pooled or arrayed formats. Pooled screens simplify the handling of large libraries with >10⁵ unique elements (*Kampmann et al., 2015*; *Morgens et al., 2016*; *Wang et al., 2015*), but require that the cellular function of interest be coupled to a simple readout, such as cell proliferation (*Han et al., 2020*; *Kory et al., 2018*) or fluorescence-activated cell sorting (FACS; *DeJesus et al., 2016*; *Potting et al., 2018*). Arrayed screens, on the other hand, are well-suited to complex readouts, such as high-content or time-lapse microscopy, and have been used to discover factors that regulate cell division (*Neumann et al., 2010*), endocytosis (*Liberali et al., 2014*), and membrane protein trafficking (*Hansen et al., 2018*; *Krumpe et al., 2012*). However, compared to pooled screens, arrayed screens are usually noisier, limited to smaller libraries (10³–10⁴), and are more technically difficult and time-consuming to implement, requiring specialized instrumentation not available to all laboratories.

To combine the strengths of the pooled screen format (library size, simplicity) and the arrayed screen format (versatility in readout), we sought to develop a molecular reporter capable of converting complex cellular processes such as protein trafficking or mislocalization into a simple, single-timepoint, intensity-based FACS readout. Such a tool would enable screening of large libraries in a pooled format without sacrificing TF the versatility and specificity required to probe more complex cellular processes.

Here we report HiLITR (High-throughput Localization Indicator with Transcriptional Readout), a molecular tool that converts protein localization or mislocalization into simple expression of a fluorescent protein. We used HiLITR in combination with a human CRISPRi library to screen for factors that regulate the trafficking of mitochondrial and ER membrane (ERM) proteins. We found that knockdown of the small ubiquitin-like modifier (SUMO) E1 ligase component SAE1 selectively destabilizes and increases the mislocalization of many mitochondrial tail-anchored (TA) proteins. Additionally, we found that knockdown of the EMC10 subunit of the ER membrane complex (EMC) increases insertion of specific TA proteins into the ERM, in opposition to the function of other EMC components.

## Results

### HiLITR is a live-cell transcriptional reporter of protein localization

To design HiLITR, we needed a mechanism to convert protein localization or mislocalization in live cells to a simple readout for pooled genetic screens. We designed two protein components – a protease (GFP-TEVp) and a transcription factor (TF), each targetable to specific subcellular locations (*Figure 1A and B*). If the protease and TF are colocalized to the same organelle (e.g., the mitochondrial membrane in *Figure 1A*), then proximity-dependent proteolysis of a protease cleavage sequence (TEVcs) in the TF's membrane anchor releases the TF, which can translocate to the nucleus and drive expression of a chosen reporter gene. If the protease and TF are not colocalized, then TF cleavage and reporter gene expression will not occur.

To maximize the dynamic range of HiLITR, we included a second 'gate' in our design – a photosensory light-, oxygen-, or voltage-sensing (LOV) domain adjacent to the TEVcs in the TF tether (*Figure 1A and B*). The LOV domain sterically blocks protease cleavage in the dark, but changes conformation to provide access under blue light illumination. HiLITR therefore acts as an AND gate, requiring both protease/TF colocalization and blue light to turn on. This two-gate design improves the dynamic range, temporal precision, and tunability compared to a one-gate design (*Kim et al., 2017*).

To test HiLITR, we generated a TF construct targeting the outer mitochondrial membrane (OMM), and protease constructs targeting the OMM, ERM, or cytosol (*Figure 1—figure supplement 1*). Transient transfection produced high background signal (*Figure 1—figure supplement 2A and B*), which was alleviated by stable integration (*Figure 1—figure supplement 2C and D* and *Figure 1—figure supplement 4A*). We improved specificity by replacing TEV protease with the approximately five fold more catalytically efficient 'ultraTEV' (S153N) (*Sanchez and Ting, 2020*) and the eLOV domain with the tighter-dark state hLOV (*Kim et al., 2017*; *Figure 1—figure supplement 2C–F*). Replacement of the Gal4 TF with a more active Gal4-VP64 fusion also improved signal (*Figure 1—figure supplement 2G and H*). Compared to our previous protein interaction-detecting tool SPARK (*Kim et al., 2017*), HiLITR detects colocalized proteins with superior sensitivity and specificity (*Figure 1—figure supplement 2I–K*). Using the optimized HiLITR constructs, we varied tool expression time, stimulation

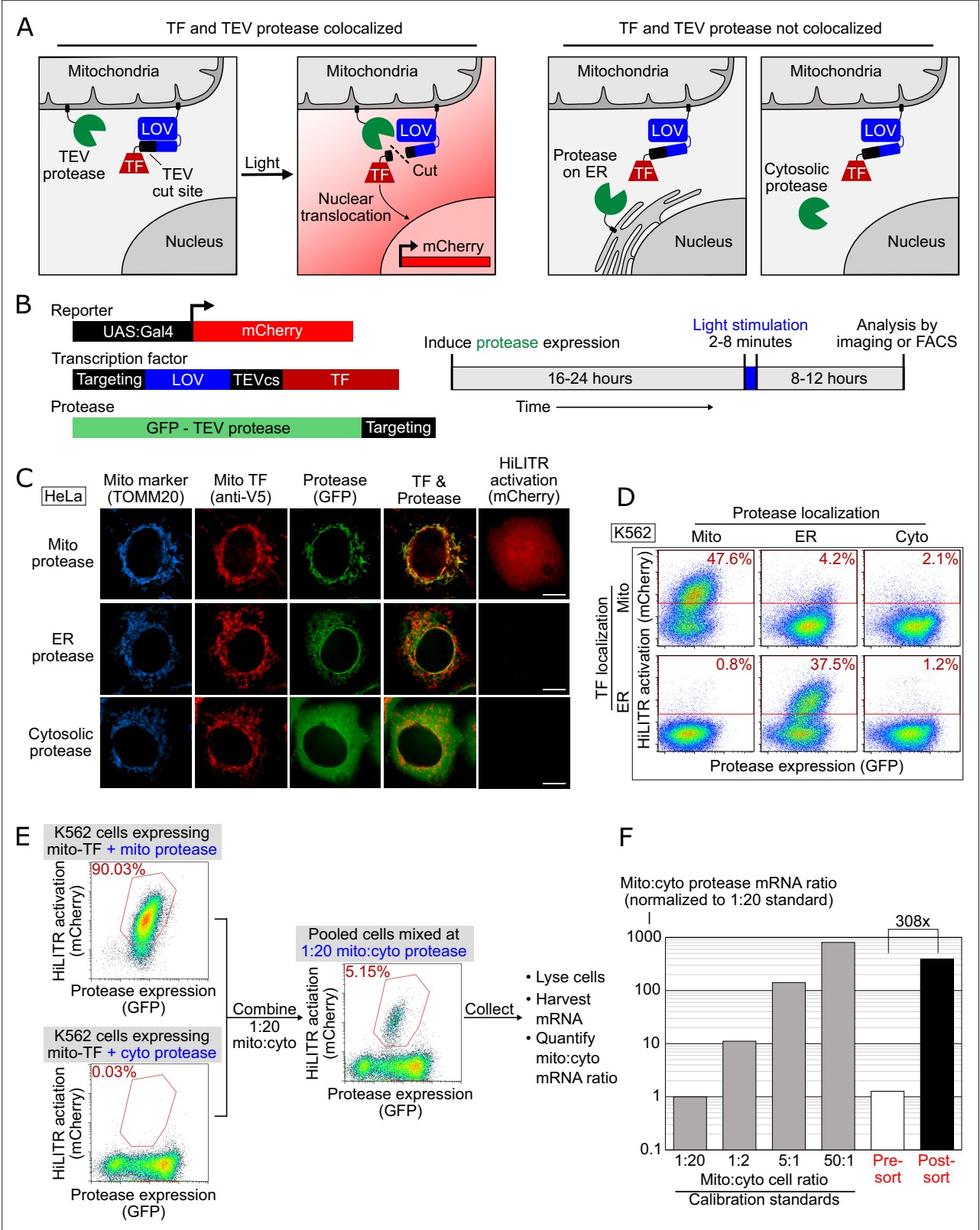

**Figure 1.** HiLITR gives transcriptional readout of protein localization in living cells. (**A**) Schematic of HiLITR. HiLITR has two components: a low-affinity protease (green) and a membrane-anchored transcription factor (TF, red). Left: when protease and TF are colocalized on the same organelle, and 450 nm blue light is supplied, the TF is released by proximity-dependent cleavage and drives reporter gene expression. Right: when protease and TF are not colocalized, HiLITR is off. (**B**) Domain structures of HiLITR components and timeline for HiLITR usage. The targeting domain is a protein or

*Figure 1 continued on next page*

*Figure 1 continued*

localization peptide that directs the TF/protease to the desired subcellular region (such as the mitochondrion in **A**, left). (**C**) Fluorescence images of HiLITR in HeLa cells. TF is on the outer mitochondrial membrane (OMM), and protease is localized to the OMM (top row), ER membrane (middle), or cytosol (bottom). mCherry is the reporter gene and TOMM20 is a mitochondrial marker. Cells were stimulated with 450 nm light for 3 min, then fixed and stained 8 hr later. Scale bars, 10 μm. (**D**) Fluorescence-activated cell sorting (FACS) plots of K562 cells expressing HiLITR. TF is on the OMM (top row) or ER membrane (bottom row), while protease localization is varied as indicated. Light stimulation was 3 min. mCherry on the y-axis reports HiLITR activation, and GFP on the x-axis reports protease expression level. Percentage of cells above the red line is quantified in each plot. (**E**) Model selection on K562 cells expressing HiLITR TF on mitochondria. Cells with mitochondrial protease (colocalized with TF) versus cytosolic protease (not colocalized with TF) were combined in a 1:20 ratio. Cells were stimulated with light for 3.5 min and sorted for high mCherry expression 8 hr later. (**F**) qPCR analysis of mito- and cyto-protease transcript from predefined, pre-sort, and post-sort cell mixtures from (**E**). Mito-protease cells were enriched 308-fold over cyto-protease cells in one round of FACS sorting. Full data in *Figure 1—source data 1*.

The online version of this article includes the following source data and figure supplement(s) for figure 1:

**Source data 1.** Source data for *Figure 1F*.

**Figure supplement 1.** Details for HiLITR constructs used in this study.

**Figure supplement 2.** Sequential optimization of HiLITR components.

**Figure supplement 3.** Optimization of HiLITR experimental parameters.

**Figure supplement 4.** Additional characterization of HiLITR constructs and cell lines.

**Figure supplement 5.** Model selection on K562 cells expressing mitochondrial transcription factor (TF) HiLITR.

**Figure supplement 5—source data 1.**

---

conditions, and reporter expression time to optimize the conditions for HiLITR use in multiple cell types (*Figure 1—figure supplement 3* and *Figure 1C*).

We further tested the versatility of HiLITR by designing a TF targeted to the ERM. By FACS in K562 cells (*Figure 1D*), we observed clear activation of ERM-TF by ERM protease, but not by OMM protease or cytosolic protease. We also saw clear activation of OMM-TF by OMM protease but not by ERM protease or cytosolic protease. The absence of cross-reactivity is striking given that mitochondria and ER form contacts in mammalian cells (*Wu et al., 2018*). Perhaps the fraction of HiLITR TF localized to these contact sites is very small compared to the total amount of TF on the OMM or ERM surface, such that the contribution of ER-mitochondria contacts to HiLITR activation is insignificant. As an additional test of specificity, we designed HiLITR constructs localized to the peroxisomal membrane, which also gave expected patterns of activation (*Figure 1—figure supplement 4F and G*).

Finally, we performed a model selection to assess our ability to enrich cells with colocalized HiLITR components from cells with non-colocalized HiLITR components. qPCR analysis showed >300- fold enrichment in a single round of FACS (*Figure 1E and F* and *Figure 1—figure supplement 5*).

## Using HiLITR in pooled CRISPRi screens to probe pathways of ER and mitochondrial membrane protein trafficking

Because HiLITR provides a simple, fluorescence intensity-based readout of protein colocalization in living cells, we sought to combine it with CRISPRi and FACS in a pooled screen to identify factors regulating the trafficking of ER and mitochondrial membrane proteins. For instance, if the HiLITR TF is targeted to the OMM via an N-terminal transmembrane anchor (signal anchor) and the HiLITR protease is localized to the OMM via a C-terminal transmembrane anchor (tail anchor, 'TA'), then any sgRNA that disrupts a gene important for TA protein targeting to the OMM should reduce HiLITR-driven reporter gene expression (*Figure 2A*). Cells with reduced HiLITR activation can be enriched by FACS and analyzed by sgRNA sequencing (*Figure 2B*).

A myriad of proteins function at the OMM and ERM, localized via signal-anchored (SA), TA, internal, and multipass transmembrane domains. Several distinct pathways orchestrate the co-translational or post-translational insertion of these proteins at the ERM (*Shao and Hegde, 2011*) and at the OMM (*Hansen and Herrmann, 2019*). Recent studies have also revealed a striking interplay between ER and mitochondrial membrane targeting pathways (*Costa et al., 2018*; *Gamerdinger et al., 2015*; *Mårtensson et al., 2019*), such as the ER-SURF pathway, in which some OMM proteins are harbored on the ERM when mitochondrial import is impaired (*Hansen et al., 2018*).

Despite our detailed and evolving picture of ER and mitochondrial protein trafficking pathways, some major gaps in understanding remain. For example, the mechanisms by which tail-anchored

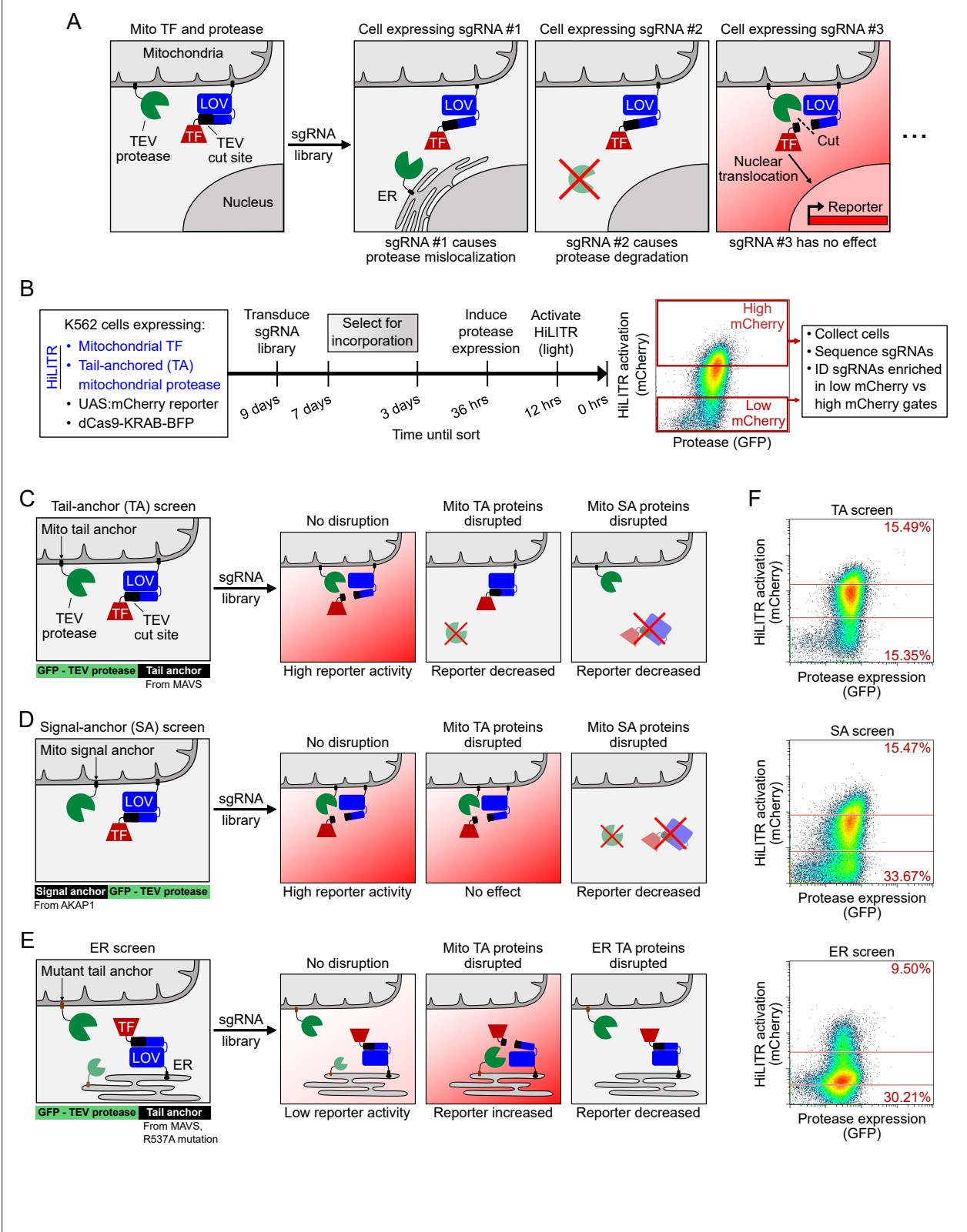

**Figure 2.** HiLITR reads out protein mislocalization or loss in CRISPRi screens. (**A**) Possible outcomes for sgRNA disruption of mitochondrial protease in cells expressing mitochondrial HiLITR transcription factor (TF) and protease. In the first example, sgRNA #1 disrupts protease localization while in the second example, sgRNA #2 reduces protease abundance. Both perturbations lead to decreased HiLITR-driven mCherry expression. (**B**) Format and timeline of CRISPRi screens. (**C–E**) Three HiLITR configurations used for CRISPRi screens. The first two (**C, D**) use mitochondria-localized TF and

*Figure 2 continued on next page*

*Figure 2 continued*

either tail-anchored (TA; **C**) or signal-anchored (SA; **D**) mitochondrial protease. The third cell line (**E**) uses ER-localized TF and a mutated tail-anchored mitochondrial protease that partitions between the outer mitochondrial membrane (OMM) and ER membrane. Examples of how HiLITR activation will be affected by various sgRNA-induced changes to protein localization are illustrated. (**F**) Fluorescence-activated cell sorting (FACS) plots showing cell populations collected and sequenced from the TA, SA, and ER CRISPRi screens from (**C**–**E**). Light stimulation times varied from 3.5 to 5 min.

The online version of this article includes the following figure supplement(s) for figure 2:

**Figure supplement 1.** Whole-genome CRISPRi screen with HiLITR readout.

mitochondrial proteins are delivered and inserted into the OMM are unclear. There are about 40 human mitochondrial TA proteins, including the apoptosis regulators BAX and BCL2 (***Wolter et al., 1997***) and mitochondrial fission proteins FIS1 (***Yoon et al., 2003***) and MFF (***Gandre-Babbe and van der Bliek, 2008***). The targeting of TA proteins presents a unique challenge because their hydrophobic transmembrane domains are translated last and must be handed off from the ribosome to appropriate chaperones or lipids before aggregation or misfolding can occur. It is known that the GET/transmembrane recognition complex (TRC) pathway, responsible for targeting of endomembrane system TA proteins to the ER, is not required for targeting of mitochondrial TA proteins (***Borgese et al., 2019***). In yeast, the HSP70 protein Sti1 (human STIP1) and Pex19 may play a role in mitochondrial TA protein targeting (***Cichocki et al., 2018***), but in vitro experiments with proteolytically shaved mitochondria show that unassisted or minimally assisted insertion may also be possible (***Kemper et al., 2008***; ***Setoguchi et al., 2006***).

We envisioned using three distinct HiLITR configurations to investigate mitochondrial TA protein insertion. In the most direct approach, impaired mitochondrial localization of a TA protease would reduce the release of a SA mitochondrial TF, diminishing HiLITR activation ('TA screen'; ***Figure 2C*** and ***Figure 1—figure supplement 1***). However, this 'loss of signal' design may enrich false-positive genes whose knockdown either nonspecifically disrupts HiLITR component expression (e.g., ribosomal proteins) or impairs trafficking of the SA TF, rather than the protease. To filter out such genes, we designed a second HiLITR screen with the same TF but an SA – rather than TA – protease ('SA screen'; ***Figure 2D*** and ***Figure 1—figure supplement 1***). Comparison of TA and SA screen results should eliminate most false positives and identify factors that selectively influence the targeting of mitochondrial TA over SA proteins.

For our third HiLITR configuration, we designed a 'gain of signal' screen based on the hypothesis that disrupted or nontargeted mitochondrial TA protease may reroute to the ERM (and subsequently to other organelles connected to the ER such as the Golgi). Thus, we expressed an ER/Golgi-targeted HiLITR TF together with a mutated mitochondrial TA protease (mTA*) that has increased propensity for mislocalization to ER/Golgi membranes ('ER screen'; ***Figure 2E*** and Figure 1—figure supplements 1 and 4D and 4E). Given the partial ER localization of this mTA* protease, we anticipated that our ER screen might also help to identify factors that regulate the targeting of native ERM proteins.

To first narrow down our list of candidate genes, we performed a whole-genome 'TA screen' with the HiLITR components shown in ***Figure 2C***. Human K562 cells expressing HiLITR and the CRISPRi repressor dCas9-BFP-KRAB were transduced with a library of sgRNAs targeting 20,000 genes, at 5 guides/gene, with 1,900 nontargeting controls. Using FACS, we collected cell populations with high mCherry expression (i.e., strong HiLITR activation) and low mCherry expression (***Figure 2—figure supplement 1A***) from two technical replicates. Next-generation sequencing of sgRNA abundance in the collected populations indicated enrichment of genes related to protein folding, membrane trafficking, and proteasome function in the low mCherry cell population (***Figure 2—figure supplement 1A and B***). As expected, we also enriched a number of gene expression-related false positives, including ribosome subunits, mediator complex subunits, and mRNA binding proteins.

Using these results, we designed an sgRNA sublibrary (***Supplementary file 2***) for simultaneous screening in the three HiLITR configurations (TA screen, SA screen, and ER screen). In total, we transduced our three clonal HiLITR K562 cell lines with 2,930 sgRNAs targeting 586 genes (5 guides/gene) plus 500 nontargeting controls. Two biological replicates were performed for each HiLITR configuration. The screens were performed as shown in ***Figure 2B***, with infection, passaging, and sorting carried out at 2,000–10,000× coverage – a higher level than standard – in order to detect subtle or partial effects. For each screen, we collected cell populations corresponding to high mCherry reporter

expression and low mCherry reporter expression (*Figure 2F*) and assessed the representation of each sgRNA between the two collected populations by next-generation sequencing.

## CRISPRi screens with three HiLITR configurations identify proteins that influence the localization of mitochondrial and ER membrane proteins

The combined results from our three HiLITR screens are shown in *Figure 3A*. Sequencing data were analyzed by CasTLE (*Morgens et al., 2016*), which assigns to each gene an effect size and an associated CasTLE score (a measure of significance, signed for effect direction). CasTLE scores in the TA and SA screens were largely concordant, reflecting the similarity between the two configurations ($R^2$ = 0.69, compared to $R^2$ = 0.28 for TA vs. ER screens and 0.37 for SA vs. ER screens). Out of 586 genes, sgRNAs against 270 of them impacted HiLITR turn-on significantly (at 10% false discovery rate [FDR]) in at least one screen (186 genes in 2+ screens). The TA screen and whole-genome screens used the same HiLITR configuration, and of the 50 most significant genes from the whole-genome screen that were included in the three-screen sublibrary, 47 were significant in the TA screen at 10% FDR, indicating good reproducibility.

To assess the validity of our screens, we first checked genes with known roles in mitochondrial and ER protein trafficking. The TRC pathway (*Schuldiner et al., 2008*; *Stefanovic and Hegde, 2007*), which handles the membrane insertion of ER TA proteins, also mishandles overexpressed mitochondrial TA proteins (*Vitali et al., 2018*). Consistent with this activity, knockdown of the TRC pathway chaperones SGTA and TRC40 significantly altered HiLITR activation in the TA and ER screens, but not the SA screen (*Figure 3A*). Other TRC pathway components also altered HiLITR activation in the TA and ER screens (*Figure 3—figure supplement 1*). The EMC, similar to the TRC pathway, handles insertion of a subset of TA proteins at the ERM (*Guna et al., 2018*). Among nine subunits tested, eight significantly altered HiLITR activation uniquely in the ER screen (Figure 6A). These results suggest that our HiLITR screens are able to recapitulate the known functions of well-characterized ER and mitochondrial membrane regulatory genes.

We then searched our data for novel genes that influence the targeting of mitochondrial TA proteins. Such proteins could be expected to have a low TA score, because sgRNA-mediated depletion of TA protease from the OMM would reduce HiLITR activation, but medium to high SA score, because SA protease would be minimally affected. A high ER score could also be expected due to a shift in mTA* protease localization from OMM to ERM, resulting in activation of ER-localized TF (*Figure 3B*). Interestingly, only a single gene met all three criteria: SAE1 (SUMO-activating enzyme 1). SAE1 is an essential protein and a member of the SAE complex (along with SAE2/UBA2), which acts as the sole SUMO E1 ligase in mammalian cells. Modification of target proteins with the small protein tag SUMO can alter protein subcellular localization (*Martin et al., 2007*; *Matunis et al., 1996*), regulate protein stability (*Desterro et al., 1998*; *Krumova et al., 2011*), and promote cellular stress response (*Golebiowski et al., 2009*). Several important mitochondrial proteins are SUMOylation targets, including Parkin (*Um and Chung, 2006*) and Drp1 (*Prudent et al., 2015*). Intriguingly, several chaperones implicated in the handling of TA proteins, including the ubiquilins (*Itakura et al., 2016*) and STIP1, are also SUMOylated (*Hendriks et al., 2018*; *Soares et al., 2013*).

As each HiLITR configuration is unlikely to be perfectly sensitive, we also looked for genes that met two of the three criteria (*Figure 3B*). Several genes involved in mitosis and cytoskeletal functions were among the hits with low TA score and mid-to-high SA score, including BORA, CCNK, REEP4, MKI67IP, and SKA1, (*Figure 3C*). The quadrant with low TA score and high ER score (*Figure 3D*) contained only a few hits, one of which was ATP6V1A, a subunit of the vacuolar ATPase, which has an important role in vesicle trafficking (*Dettmer et al., 2006*). Additional hits and pathways are discussed in *Figure 3—figure supplements 2–4*.

To check the robustness of our hits, we individually validated single sgRNAs in our HiLITR cell lines (Figure 3—figure supplements 2A and C and 3). In addition to HiLITR activation, we also quantified expression level of the GFP-tagged protease as mistargeting may result in protein degradation and reduction of GFP signal (*Figure 3—figure supplement 2B* and 2D). We selected SAE1 (*Figure 4*) and seven additional hits from *Figure 3D and E* for validation. Four of these hits (SAE1, CCNK, SKA1, and ATP6V1A) were validated (*Figure 4* and *Figure 3—figure supplement 2E*), and we found by imaging that knockdown of SKA1 or ATP6V1A increased the fraction of GFP-mTA* protease mislocalized to

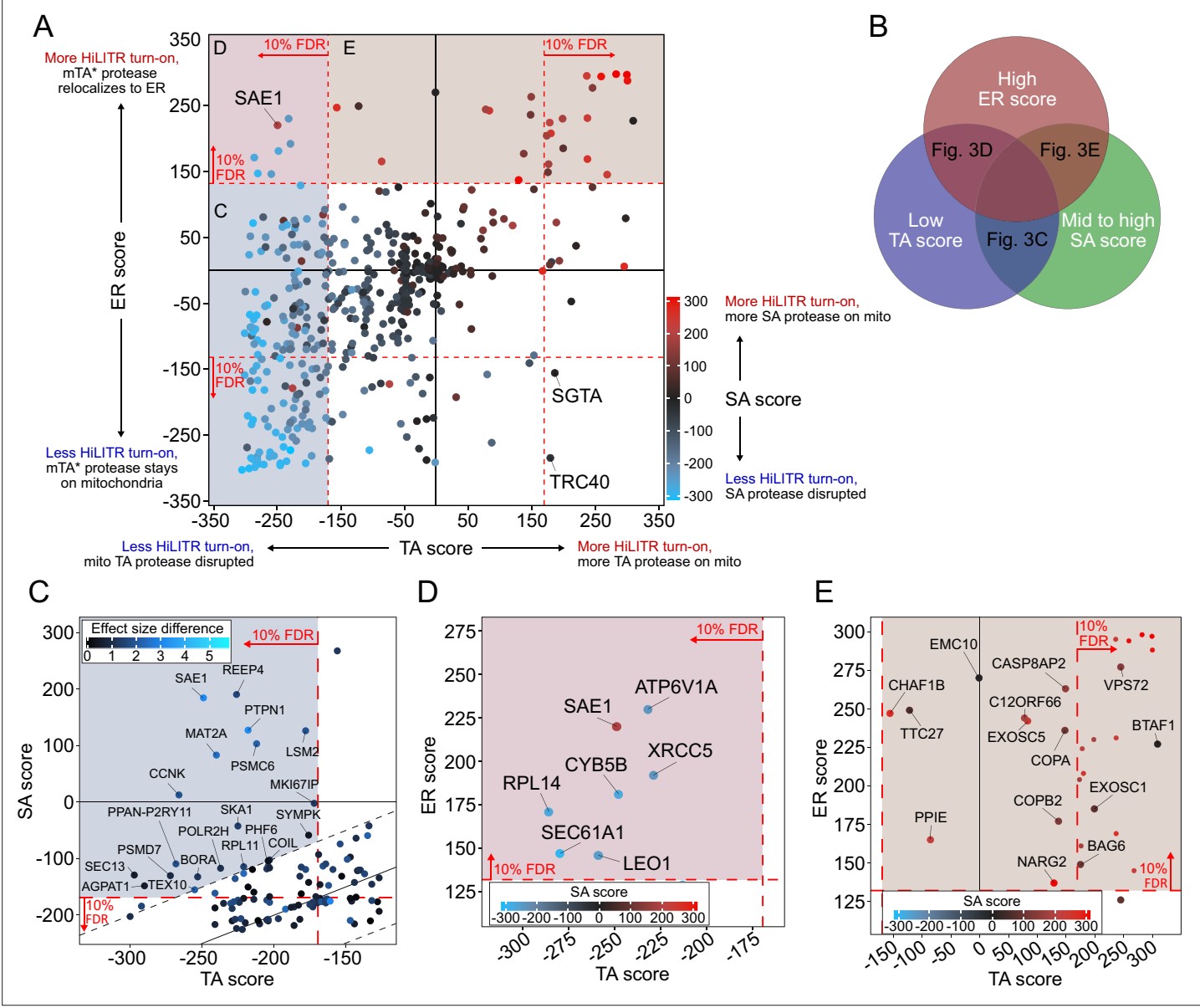

**Figure 3.** CRISPRi screen with HiLITR readout identifies proteins that influence the localization of mitochondrial membrane and ER membrane proteins. (**A**) CasTLE plot showing combined results from tail-anchored (TA), signal-anchored (SA), and ER CRISPRi screens. x-axis plots the TA screen score (lower when the mito TA protease from *Figure 2C* is disrupted) and y-axis plots the ER screen score (higher when the mito mTA* protease from *Figure 2D* relocalizes to the ER membrane). Points are color-coded according to score in the SA screen, with red denoting less disruption of the mito SA protease from *Figure 2C*. (**B**) Venn diagram showing that proteins regulating the targeting of mitochondrial TA proteins may exhibit some combination of low TA score, mid to high SA score, and high ER score. (**C**) Zoom-in of proteins with low TA score and medium-high SA score (replotted from **A**). Points are colored according to absolute difference in effect size in TA vs. SA screen. Dashed black lines enclose the 90% interquantile range for difference between TA and SA score. (**D**) Zoom-in of proteins with low TA score and high ER score, corresponding to maroon shaded region in (**A**). (**E**) Zoom-in of proteins with high ER score and medium-high SA score, corresponding to brown shaded region in (**A**). Unlabeled points showed significant increases in HiLITR activation (at 10% FDR) in all three screens and are likely to be nonspecific hits.

The online version of this article includes the following source data and figure supplement(s) for figure 3:

**Figure supplement 1.** Retesting of transmembrane recognition complex (TRC)/GET pathway genes with HiLITR.

**Figure supplement 1—source data 1.**

**Figure supplement 1—source data 2.**

**Figure supplement 2.** Retesting individual sgRNAs from the three HiLITR configuration CRISPRi screen.

**Figure supplement 3.** Retesting individual sgRNAs in polyclonal cell lines.

**Figure supplement 4.** Analysis of other pathways in CRISPRi screening data.

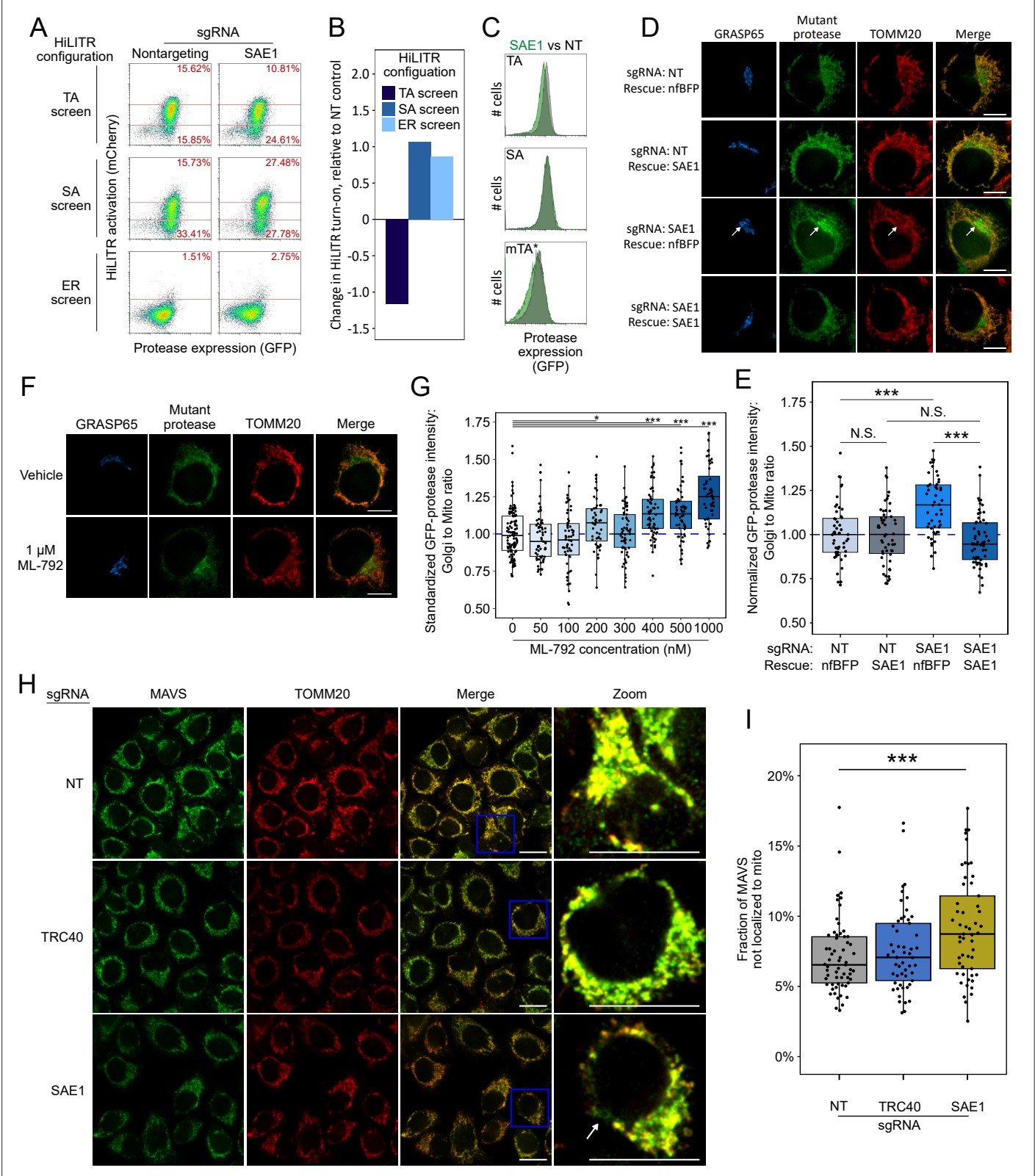

**Figure 4.** SAE1 knockdown disrupts localization and abundance of mitochondrial tail-anchored (TA) proteins. (**A**) SAE1 knockdown by CRISPRi reduces HiLITR activation in TA screen configuration, while increasing HiLITR activation in SA and ER screen configurations. (**B**) Quantitation of data in (**A**). Log2-transformed ratio of high mCherry to low mCherry cells was calculated for each plot and normalized to that of nontargeting (NT) sgRNA control. (**C**) Expression levels of GFP-tagged mitochondrial proteases from samples in (**A**). Green: SAE1 knockdown cells; gray: control cells with NT guide. (**D**) SAE1

*Figure 4 continued on next page*

*Figure 4 continued*

knockdown increases mislocalization of the GFP-tagged mutant TA (mTA*) protease from mitochondria to Golgi. HeLa cells expressing mTA* protease and dCas9-KRAB were infected with SAE1 sgRNA or NT control for 9 days. In rows 2 and 4, SAE1 knockdown was rescued by overexpression of sgRNA-resistant SAE1. nfBFP: non-fluorescent BFP. Mitochondria and Golgi are visualized with anti-TOMM20 and anti-GRASP65 antibodies, respectively. In SAE1 knockdown without rescue (third row), mutant GFP-protease accumulates in Golgi (white arrow). Scale bars, 10 μm. (**E**) Quantitation of data in (**D**) along with ~20 additional fields of view (n = ~50 cells per condition). The value plotted is the mean intensity of GFP-protease signal colocalized with Golgi divided by mean signal colocalized with mitochondria. N.S.: not significant, ***$p<0.001$, Student's t-test. Full data in *Figure 4—source data 1*. (**F**) Chemical inhibition of SAE1's SUMOylation activity increases mislocalization of the GFP-tagged mTA* protease to the Golgi. HeLa cells were treated with SUMO E1-ligase inhibitor ML-792 for 6 days before expression of mTA* protease for 1 day. Localization of the GFP-tagged mutant protease was compared with respect to mitochondrial and Golgi markers. Scale bars, 10 μm. (**G**) Quantitation of the data in (**F**), with six additional concentrations of ML-792 inhibitor. ~20 fields of view (n = ~50 cells) were imaged per condition. *$p<0.05$, ***$p<0.001$, Student's t-test. Full data in *Figure 4—source data 2*. (**H**) SAE1 knockdown increases the fraction of endogenous mitochondrial TA protein MAVS that is mislocalized. HeLa cells were infected with nontargeting control or sgRNA against TRC40 or SAE1 for 9 days. Endogenous MAVS and the mitochondrial marker TOMM20 were visualized by immunostaining. Zooms are contrast-enhanced. White arrow points to MAVS signal in a non-mitochondrial region. Scale bars, 20 μm. (**I**) Quantitation of data in (**I**) along with approximately five additional fields of view (n = ~ 60 cells per condition). ***$p<0.001$, Student's t-test. Full data in *Figure 4—source data 3*.

The online version of this article includes the following source data and figure supplement(s) for figure 4:

**Source data 1.** Source data for Figure 4E.

**Source data 2.** Source data for Figure 4G.

**Source data 3.** Source data for Figure 4I.

**Figure supplement 1.** Additional data on SAE1.

**Figure supplement 1—source data 1.** Source data for Figure 4 - figure supplement 1E.

**Figure supplement 1—source data 2.** Source data for Figure 4 - figure supplement 1J.

the Golgi (*Figure 3—figure supplement 2F* and 2G; a consequence of anterograde trafficking after mistargeting to the ER).

## SAE1 knockdown disrupts localization and abundance of many mitochondrial TA proteins

Validation with individual sgRNAs against SAE1 in HiLITR cell lines recapitulated the results of the CRISPRi screens (*Figure 4A and B* and *Figure 4—figure supplement 1A* and B). In addition, we observed that SAE1 knockdown specifically reduced the abundance of GFP-tagged mitochondrial TA protease, but not GFP-tagged mitochondrial SA protease (*Figure 4C* and *Figure 4—figure supplement 1C*). This may be because a significant fraction of mitochondrial TA protease that fails to target to the OMM is destabilized and degraded. We also used confocal microscopy to analyze the subcellular localization of GFP-tagged mitochondrial mTA* protease. We found that knockdown of SAE1 increases the mislocalization of GFP-mTA* protease to ER/Golgi compartments, as measured by the ratio of GFP overlapping with Golgi versus mitochondrial markers. The effect was rescued by overexpression of sgRNA-resistant SAE1 gene (*Figure 4D and E* and *Figure 4—figure supplement 1D* and E).

Knockdown of SAE1 might impair topogenesis of the TA mitochondrial proteases through SUMOylation activity or through undetermined nonenzymatic binding interactions. To distinguish between these possibilities, we examined the effects of the small-molecule inhibitor ML-792, which inhibits global SUMOylation with an EC50 of 19 nM (*Huang et al., 2018*). Inhibition of SUMOylation in K562 cells for just 2 days reproduced the effects of SAE1 knockdown on HiLITR activation in the TA, SA, and ER screen cell lines (*Figure 4—figure supplement 1F–H*). Furthermore, in HeLa cells expressing the GFP-tagged mitochondrial mTA* protease, inhibition of SUMOylation with ML-792 in excess of 400 nM increased mislocalization of GFP-mTA* protease to Golgi (*Figure 4F and G*).

We next examined the effect of SAE1 knockdown on endogenous rather than recombinant mitochondrial proteins. First, we used fluorescence microscopy to assess changes in protein localization upon SAE1 knockdown. In control cells expressing nontargeting sgRNA, the TA mitochondrial protein MAVS almost completely localizes to the mitochondrion (*Figure 4H*), although some non-mitochondrial MAVS appears in punctate structures that may correspond to peroxisomes (*Dixit et al., 2010*). Knockdown of the ER-specific chaperone TRC40 did not alter the extent of colocalization between MAVS and the mitochondrial marker TOMM20. In contrast, knockdown of SAE1 significantly increased the

non-mitochondrial fraction of MAVS (*Figure 4I*). In these cells, we observed non-mitochondrial MAVS in fibrous, perinuclear structures (*Figure 4H*, zoom) distinct from the small puncta observed in control samples. In contrast to MAVS, the localization of endogenous AKAP1, an SA mitochondrial protein, was unaffected by knockdown of SAE1 (*Figure 4—figure supplement 1I* and J).

We also used western blotting to assess the abundance of specific endogenous mitochondrial TA proteins. HeLa cells expressing dCas9-KRAB were infected with a nontargeting sgRNA or an sgRNA against SAE1. After 9 days of guide expression, we harvested cells and measured the abundance of endogenous mitochondrial proteins, using GAPDH as a loading control because it is known to not be SUMOylated (*Huang et al., 2018*). Of three mitochondrial TA proteins tested (MAVS, SYNJ2BP, and FIS1), two showed significant depletion upon knockdown of SAE1 (*Figure 5A and B* and *Figure 5—figure supplement 1*). In contrast, three non-TA mitochondrial proteins with diverse targeting signals (COX4, VDAC1, and AKAP1) all showed no reduction in protein abundance upon SAE1 knockdown (*Figure 5A and B* and *Figure 5—figure supplement 1*). Overexpression of sgRNA-resistant SAE1 partially restored levels of the mitochondrial TA proteins (*Figure 5—figure supplement 2*).

To examine the mitochondrial proteome in a more global and quantitative manner, we performed mass spectrometry-based proteomic analysis on SAE1 knockdown HeLa samples. HeLa cells overex-pressing nonfluorescent BFP control or sgRNA-resistant SAE1 were each transduced with nontargeting sgRNA or sgRNA against SAE1 for 9 days. These samples were harvested in triplicate and whole-cell lysates were analyzed by mass spectrometry (*Figure 5—figure supplement 3A*). We found that over-expression of SAE1 in cells expressing nontargeting control guide produced no significant changes in the overall proteome (*Figure 5—figure supplement 3B*). By contrast, because SAE1 is essential, its knockdown affected a large swath of the human proteome (*Figure 5—figure supplement 3C*), with 10.8% of proteins changing in abundance by 1.25-fold or more (see *Supplementary file 3* for complete proteomic data). Surprisingly, knockdown of SAE1 also substantially increased the abun-dance of the mitochondrial proteome relative to the non-mitochondrial proteome (*Figure 5—figure supplement 3C*). To examine the mitochondrial proteome specifically, we normalized the abundance of the 889 detected mitochondrial proteins to their collective mean (*Figure 5—figure supplement 4A* and B). Knockdown of SAE1 resulted in relative depletion of seven TA mitochondrial proteins while only two were enriched (*Figure 5C* and *Figure 5—figure supplement 4A*). Rescue of SAE1 reversed these trends (*Figure 5—figure supplement 4B* and C). Using data from both the knockdown and rescue samples, the abundance of a majority of the detected mitochondrial TA proteins positively correlated with that of SAE1, including FIS1 from the western blot data (*Figure 5D*). In contrast, only a quarter of the overall mitochondrial proteome positively correlated with SAE1, and neither SA proteins nor other transmembrane proteins of the OMM significantly deviated from this trend (*Figure 5D*). When we performed the same analysis on the ER proteome, neither TA nor SA proteins of the ER significantly deviated from the rest of the ER proteome (*Figure 5—figure supplement 5*).

Taken together, our results suggest that SAE1 knockdown impairs the targeting and triggers the degradation of MAVS and several other endogenous mitochondrial TA proteins, but not of other transmembrane mitochondrial or ER proteins.

## EMC10 is an EMC component with a distinct regulatory effect on ERM proteins

In our ER screen, the HiLITR TF is localized to the ERM, and the mTA* protease is distributed between the OMM and ERM (*Figure 1—figure supplement 4E*). Therefore, it is possible for the ER screen to also identify regulators of ER TA proteins, whose knockdown could decrease colocalization of the mTA* protease and ER-TF, reducing HiLITR activation. We closely examined hits that gave large CasTLE scores in the ER screen. Several of the highest scoring genes were components of the EMC, which mediates proper insertion of both multipass transmembrane proteins (*Chitwood et al., 2018*; *Shurtleff et al., 2018*) and a subset of TA proteins (*Guna et al., 2018*) into the ERM.

Seven of the nine EMC subunits included in our screen reduced HiLITR activity in the ER screen but had no effect in the TA or SA screens (*Figure 6A*; EMC subunits 1/2/3/4/6/7/8). This suggests that our mTA* protease is a client of EMC, while the mitochondrial TA and SA proteases do not interact with the EMC. In contrast to the other subunits, EMC10 strongly *increased* HiLITR activity in the ER screen (*Figure 6A*). EMC10 is less well-conserved than other EMC proteins (*Wideman, 2015*), does not cluster with core components in genetic interaction mapping (*Jonikas et al., 2009*), and is dispensable

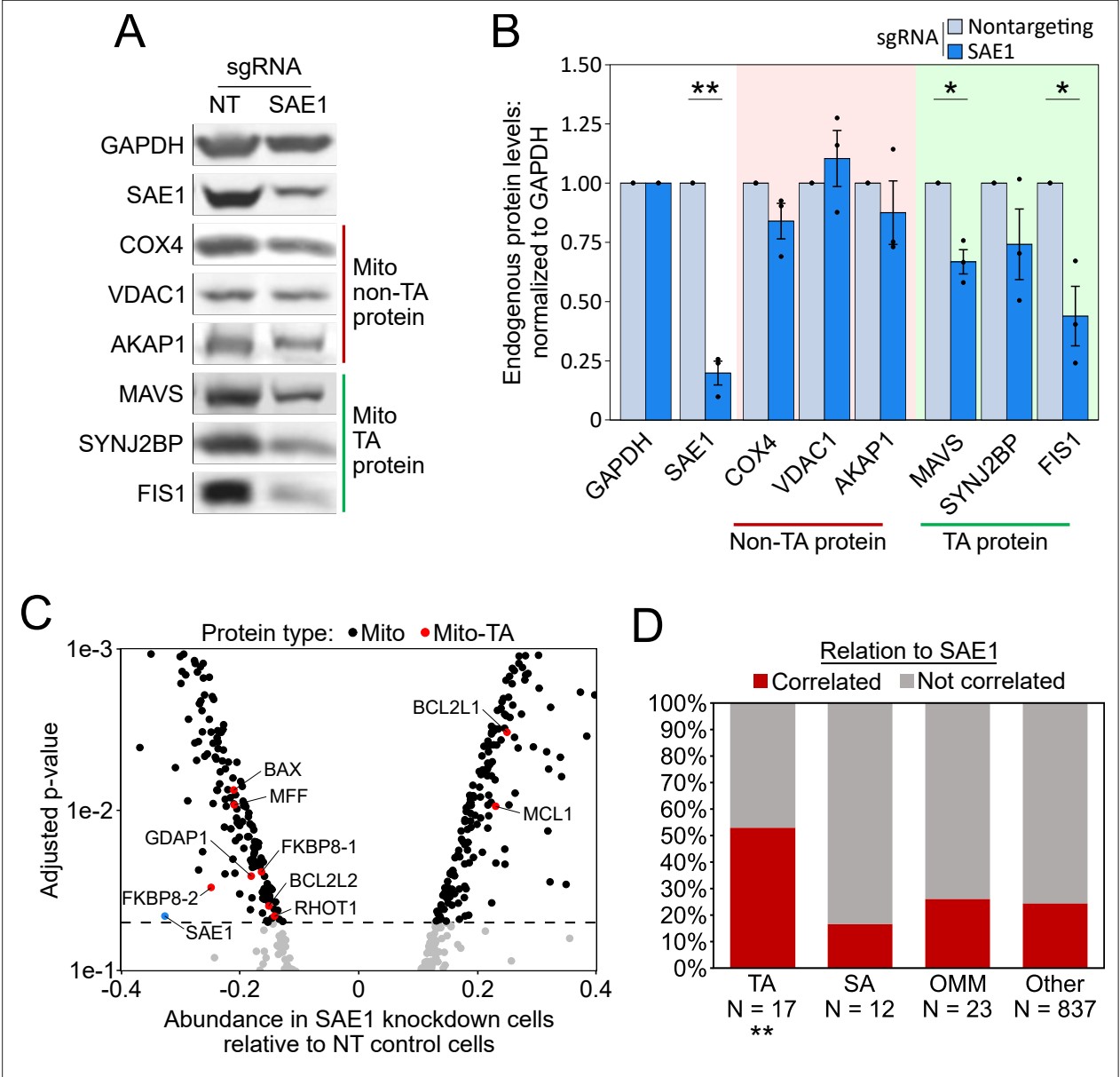

**Figure 5.** SAE1 knockdown reduces the abundance of many endogenous mitochondrial tail-anchored (TA) proteins. (**A**) HeLa cells infected with SAE1 sgRNA or nontargeting control for 9 days were analyzed by western blot. Three TA mitochondrial proteins (MAVS, SYNJ2BP, FIS1) were analyzed in addition to three non-TA mitochondrial proteins (COX4, VDAC1, AKAP1). Uncropped blots in *Figure 5—figure supplement 1*. (**B**) Quantification of data in (**A**) along with two additional biological replicates per condition. Error bars = SEM. *p<0.05, **p<0.01, Student's t-test. Full data in *Figure 5—source data 1*. (**C**) Proteomic analysis of endogenous mitochondrial protein abundance in whole-cell lysate from SAE1 knockdown HeLa cells. Enrichment scores (abundance in SAE1 knockdown samples relative to abundance in nontargeting control samples; same samples as in *Figure 5—figure supplement 3*) were normalized to the mean mitochondrial protein abundance. Dashed line, p=0.05. Full volcano plot in *Figure 5—figure supplement 4*. (**D**) Percentage of different protein classes whose abundance positively correlates with that of SAE1 (red). TA: mitochondrial tail-anchored proteins; SA: mitochondrial signal-anchored proteins; OMM: other transmembrane outer mitochondrial membrane proteins; other: all other mitochondrial proteins. **p<0.01, chi-square test against 'other' mitochondrial proteins.

The online version of this article includes the following source data and figure supplement(s) for figure 5:

**Source data 1.** Source data for Figure 5B.

**Figure supplement 1.** Uncropped western blots used to make *Figure 5A and B*.

**Figure supplement 2.** Western blots of SAE1 knockdown and rescue.

**Figure supplement 3.** Whole-proteome profiling data.

**Figure supplement 4.** Mitochondrial proteome data normalized to mean mitochondrial protein abundance.

*Figure 5 continued on next page*

*Figure 5 continued*

**Figure supplement 5.** ER proteome data normalized to mean ER protein abundance.

for complex stability (*Volkmar et al., 2019*). EMC10 forms contacts with EMC1 and EMC7 in the ER lumen (*O'Donnell et al., 2020*), and mutations at the EMC1/7 interface strongly increase the level of a reporter based on the canonical TA EMC client SQS (*Miller-Vedam et al., 2020*). Therefore, we hypothesized that EMC10 may antagonize or regulate the activity of the EMC, such that its depletion increases rather than decreases the insertion of client TA proteins.

We began by validating the HiLITR screen results of EMC4, EMC8, and EMC10. We chose EMC4 because it is part of the main cavity (along with EMC3/6) that mediates insertion of EMC substrates (*Bai et al., 2020*; *Pleiner et al., 2020*), but its depletion does not destabilize the rest of the EMC (*Volkmar et al., 2019*). Compared to two nontargeting controls, EMC4 and EMC8 knockdown both decreased HiLITR activity specifically in the ER screen configuration, while two guides against EMC10 both increased HiLITR activity (*Figure 6B* and *Figure 6—figure supplement 1A*), consistent with the results of our screen.

We next used fluorescence microscopy to assess the distribution of GFP-tagged mTA* protease in HeLa cells (*Figure 6C and D*). While knockdown of EMC4 and EMC8 both decreased the colocalization of GFP-mTA* protease with the Golgi, the guides against EMC10 increased the amount of GFP-mTA* protease at the Golgi. This suggests that EMC10 knockdown increases insertion of mTA* protease at the ERM relative to the OMM.

Although the mTA* protease appears to be a client of the EMC, it is not derived from a canonical ER TA protein. To query a more representative construct, we generated two new proteases targeted to the ERM via fusion to the native ER TA protein SQS or to its transmembrane domain (*Figure 1—figure supplement 1* and *Figure 6—figure supplement 1B*). In K562 cells expressing the HiLITR ER-TF, knockdown of EMC10 increased HiLITR activation with full-length SQS protease (*Figure 6—figure supplement 1C–F*). We next tested the effects of EMC10 knockdown on endogenous SQS. HeLa cells were infected with guides against EMC4 or EMC10, and cells were harvested after 9 days of sgRNA expression. Western blotting confirmed knockdown of EMC4 and EMC10 (*Figure 6E* and *Figure 6—figure supplement 2*). Neither EMC4 nor EMC10 knockdown altered the levels of the ER-translocon-dependent lumen protein calreticulin (CALR) or the TRC-dependent TA protein VTI1B. In contrast, EMC4 knockdown significantly decreased the level of SQS, while EMC10 knockdown significantly increased SQS levels (*Figure 6E and F* and *Figure 6—figure supplement 2*).

## Discussion

With the advent of CRISPR-based gene perturbation screens and continued improvements to next-generation sequencing, large-scale functional genomics studies have become simpler, faster, and more cost-effective, particularly for pooled-format screens using conventional equipment and reagents. Presently, the biology that can be accessed by pooled-format screens is most limited by our ability to couple cellular processes of interest to a simple and robust readouts. This presents an opportunity for molecular reporter development to contribute to the field of functional genomics.

Several recent studies have combined pooled cell culturing with automated high-content microscopy using in-place sequencing (*Feldman et al., 2019*; *Wang et al., 2019*), arrayed imaging (*Wheeler et al., 2020*), or photoinducible reporters (*Kanfer et al., 2021*; *Yan et al., 2021*) to identify hits. Such approaches simplify cell culturing and collection, but identifying phenotypic hits still requires time-consuming microscopy and computational analysis. In this work, we have developed an alternative approach to the study of protein localization with pooled screens. HiLITR is a genetically encoded tool that provides light-gated readout of protein localization, enabling fast, genome-wide screening with high coverage, while dispensing with the need for specialized equipment or software.

Protein complementation assays (PCAs), such as split GFP (*Cabantous et al., 2005*), can also be used as readouts in high-throughput assays. Compared to PCAs, HiLITR provides signal amplification, which may improve sensitivity toward weak or rare effects. The customizable HiLITR reporter is also not limited to fluorescent protein production.

We envision a wide range of possible HiLITR applications in future studies. Because HiLITR is modular, it can be designed for applications at other organelles. HiLITR is also not limited to single-pass

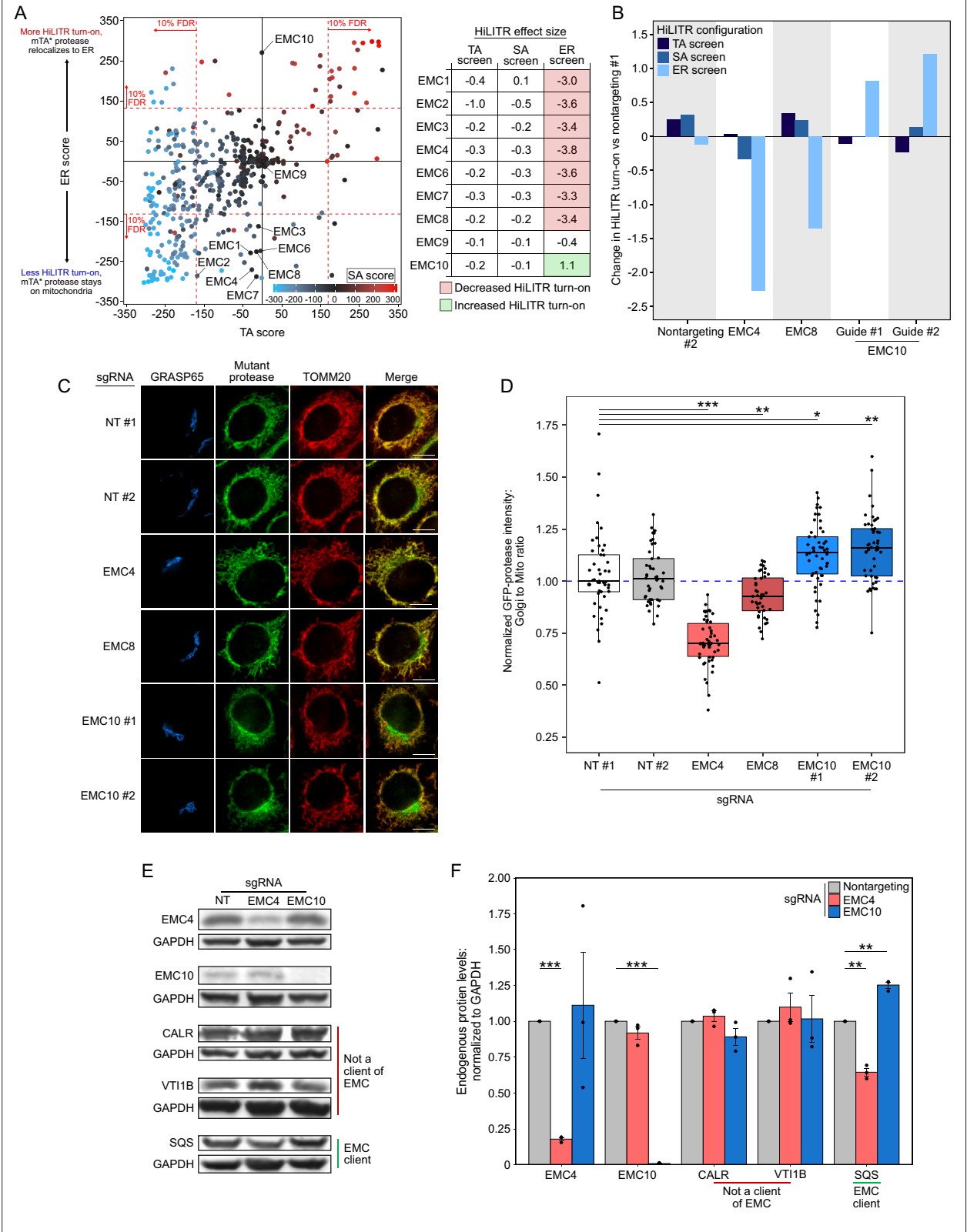

**Figure 6.** EMC10 has opposite regulatory effect on ER tail-anchored (TA) proteins as other ER membrane complex (EMC) subunits. (**A**) Locations of 9 of 10 EMC components in the 3-CRISPRi screen CasTLE plot from *Figure 3A*. EMC5 was not included in the screen. In the table at right, corresponding effect sizes from each screen are shown. (**B**) Quantitation of the effect of individual EMC subunit (4, 8, and 10) knockdown in the TA, signal-anchored (SA), and ER HiLITR cell lines. Fluorescence-activated cell sorting (FACS) data shown in *Figure 6—figure supplement 1*. (**C**) Knockdown of EMC10

*Figure 6 continued on next page*

*Figure 6 continued*

increases, while knockdown of EMC4 or EMC8 decreases, the mislocalization of GFP-tagged mTA* protease from mitochondria to Golgi in HeLa cells. Golgi and mitochondria are detected with anti-GRASP65 and anti-TOMM20 antibodies. Scale bar, 10 µm. (**D**) Quantification of data in (**C**) along with ~20 additional fields of view per condition (~50 cells per sample). *p<0.05, **p<0.01, ***p<0.001, Student's t-test. Full data in *Figure 6—source data 1*. (**E**) Knockdown of EMC subunits has different effects on endogenous EMC client protein SQS. HeLa cells expressing the indicated sgRNAs (EMC10 sgRNA #2) for 9 days were analyzed by western blot to detect the endogenous TA EMC client protein SQS as well as two non-client proteins (ER lumen protein CALR and ER TA protein VTI1B). Uncropped blots in *Figure 6—figure supplement 2*. (**F**) Quantification of data in (**E**) along with two additional biological replicates per condition. Error bars = SEM. **p<0.01, ***p<0.001, Student's t-test. Full data in *Figure 6—source data 2*.

The online version of this article includes the following source data and figure supplement(s) for figure 6:

**Source data 1.** Source data for Figure 6D.

**Source data 2.** Source data for Figure 6F.

**Figure supplement 1.** Additional HiLITR analysis related to the ER membrane complex (EMC).

**Figure supplement 2.** Uncropped western blots used to make *Figure 6E and F*.

transmembrane proteins. Provided that the TF is robustly excluded from the nucleus and both TF and protease are cytosol-exposed with compatible geometries, HiLITR can be applied to study the localization of multipass, peripheral, or potentially even cytosolic proteins. As demonstrated here, some iteration may be necessary to optimize HiLITR geometry and dynamic range.

Though not exploited here, the light-gating of HiLITR could be applied to time-resolved detection of transient changes in protein localization, such as protein translocation in response to internal or external cues. HiLITR could also be formatted for use in other high-throughput assays, such as screens of chemical libraries.

While powerful, HiLITR does have limitations. Loss-of-activation screens produce false positives that nonspecifically decrease the expression of HiLITR components. Repeated applications of HiLITR by the scientific community may generate lists of recurring false positives, but false positives can also be filtered through the use of matched counterscreens (e.g., the SA screen vs. the TA screen used here). Gain-of-activation screens are likely to produce fewer false positives, but they must be designed with prior knowledge of how perturbation will alter construct localization. We also recommend cautious interpretation of hits that produce strong growth defects, for which we have observed generally lower reproducibility. Finally, because HiLITR uses recombinant, chimeric proteins and incorporates signal amplification, changes in HiLITR activation might not perfectly match the magnitude or scope of changes to the regulation of endogenous proteins. Larger HiLITR effect sizes will align with larger functional outputs, but unless conditions can be calibrated to treatments of known effect size, HiLITR should not be used to make specific claims about the magnitude of an effect. Follow-up validation on endogenous proteins is necessary for full confidence in the specificity of experimental observations.

In this study, we combined HiLITR with CRISPRi to identify genes involved in the trafficking of mitochondrial and ERM proteins. Using a pooled format, we performed a whole-genome screen followed by three smaller-scale screens, focusing on the identification of genes that specifically perturb the trafficking of mitochondrial TA proteins. We identified SAE1, an essential member of the mammalian SUMO E1 ligase. Knockdown of SAE1 specifically decreased the abundance of the endogenous mitochondrial TA proteins, including MAVS and FIS1. Knockdown of SAE1 also promoted mislocalization of endogenous MAVS. The effects of SAE1 knockdown were phenocopied by chemically inhibiting SUMOylation.

While the effect of SAE1 knockdown appears to be specific to mitochondrial TA proteins, we hypothesize that the perturbation is most likely indirect. Both STIP1 and the ubiquilins show evidence of SUMOylation (*Hendriks et al., 2018*), and it is possible that SUMOylation of these or other chaperones alters their activity, client specificity, or subcellular distribution. Alternatively, the effects of SAE1 knockdown may be mediated through changes to cell proliferation. SAE1 was one of several hits involved in regulation of the cytoskeleton and of mitosis. In our proteomics experiment, knockdown of SAE1 globally upregulated most mitochondrial proteins (relative to non-mitochondrial proteins), with the exception of TA proteins. Perhaps cellular and mitochondrial proliferation become uncoupled upon SAE1 knockdown. TA proteins, which must be post-translationally inserted into the OMM, may be less influenced by the factors that alter mitochondrial proliferation relative to cellular proliferation.

In our ER screen, we observed a novel and unexpected consequence of EMC10 knockdown. While loss of other EMC subunits decreased the ER localization of our mutant TA protease, EMC10 knockdown produced the opposite effect, which we confirmed by immunofluorescence and FACS. We further showed that knockdown of EMC10 increased abundance of the endogenous ER TA protein SQS. EMC10 localizes to the ER lumen, so it is not directly involved in EMC client recognition. However, the EMC has been shown to occupy two distinct conformations that differ in accessibility of the insertase transmembrane cavity, and lumenal domain rotation is observed between these states. Additionally, disruption of the lumenal interface between EMC1/7 increases the level an SQS-based reporter construct, and it has been suggested that permissiveness of the EMC might be adjusted to regulate levels of SQS as the cell responds to changes in demand for sterol synthesis (*Miller-Vedam et al., 2020*). We propose that EMC10 plays a role in this regulation. EMC10 sits below the insertase cavity on the lumenal side and engages with both EMC1/7. Knockdown of EMC10 in our experiments produces results concordant with mutation of the EMC1/7 interface (*Miller-Vedam et al., 2020*). We therefore speculate that EMC10 stabilizes the EMC1/7 junction and the closed insertase conformation, and that EMC10 may dissociate from the EMC to promote increased insertase activity. In addition to its role as an insertase, the EMC also serves as a holdase chaperone to favor proper orientation of multipass transmembrane proteins of the endomembrane system (*Chitwood et al., 2018*). It remains to be shown whether EMC10 plays a regulatory role in this function of the EMC.

# Materials and methods

**Key resources table**

| Reagent type (species) or resource | Designation | Source or reference | Identifiers | Additional information |
|---|---|---|---|---|
| Cell line (human) | HEK293T | ATCC | Cat# CRL-3216; RRID:CVCL_0063 | |
| Cell line (human) | K562 | ATCC | Cat# CCL-243; RRID:CVCL_0004 | |
| Cell line (human) | HeLa | *Hein et al., 2015* | RRID:CVCL_1922 | |
| Antibody | Anti-V5 (Mouse monoclonal) | Invitrogen | Cat# R960; RRID:AB_2556564 | Immunofluorescence (1:1000) |
| Antibody | Anti-TOMM20 (Rabbit monoclonal) | Abcam | Cat# ab186735; RRID:AB_2889972 | Immunofluorescence (1:500) |
| Antibody | Anti-GRASP65 (Mouse monoclonal) | Santa Cruz | Cat# sc-374423; RRID:AB_10991322 | Immunofluorescence (1:500) |
| Antibody | Anti-CANX (Rabbit polyclonal) | Thermo Fisher | Cat# PA5-34754; RRID:AB_2552106 | Immunofluorescence (1:500) |
| Antibody | Anti-PEX14 (Rabbit polyclonal) | Proteintech | Cat# 10594-1-AP; RRID:AB_2252194 | Immunofluorescence (1:500) |
| Antibody | Anti-RCN2 (Rabbit polyclonal) | Thermo Fisher | Cat# PA5-56542; RRID:AB_2646431 | Immunofluorescence (1:500) |
| Antibody | Anti-GAPDH (Mouse monoclonal) | Santa Cruz | Cat# sc-32233; RRID:AB_627679 | Western blot (1:4500) |
| Antibody | Anti-SAE1 (Rabbit polyclonal) | Sigma | Cat# SAB4500028; RRID:AB_10742679 | Western blot (1:500) |
| Antibody | Anti-COX4 (Rabbit polyclonal) | Abcam | Cat# ab16056; RRID:AB_443304 | Western blot (1:1000) |
| Antibody | Anti-VDAC1 (Mouse monoclonal) | Abcam | Cat# ab14734; RRID:AB_443084 | Western blot (1:500) |
| Antibody | Anti-AKAP1 (Mouse monoclonal) | Santa Cruz | Cat# sc-135824; RRID:AB_2225573 | Immunofluorescence (1:200); western blot (1:500) |
| Antibody | Anti-MAVS (Mouse monoclonal) | Santa Cruz | Cat# sc-166583; RRID:AB_2012300 | Immunofluorescence (1:200); western blot (1:250) |

*Continued on next page*

*Continued*

| Reagent type (species) or resource | Designation | Source or reference | Identifiers | Additional information |
|---|---|---|---|---|
| Antibody | Anti-SYNJ2BP (Rabbit polyclonal) | Sigma | Cat# HPA000866; RRID:AB_2276678 | Western blot (1:500) |
| Antibody | Anti-FIS1 (Rabbit polyclonal) | Thermo Fisher | Cat# 10956–1-AP; RRID:AB_2102532 | Western blot (1:1000) |
| Antibody | Anti-EMC4 (Rabbit monoclonal) | Abcam | Cat# ab184162; RRID:AB_2801471 | Western blot (1:1000) |
| Antibody | Anti-EMC10 (Rabbit monoclonal) | Abcam | Cat# ab180148; RRID:AB_2889936 | Western blot (1:500) |
| Antibody | Anti-CALR (Rabbit polyclonal) | Thermo Fisher | Cat# PA3900; RRID:AB_325990 | Western blot (1:500) |
| Antibody | Anti-VTI1B (Rabbit monoclonal) | Abcam | Cat# ab184170; RRID:AB_2889935 | Western blot (1:250) |
| Antibody | Anti-SQS (Rabbit monoclonal) | Abcam | Cat# ab195046; RRID:AB_2860018 | Western blot (1:500) |
| Antibody | Anti-Mouse Alexa Fluor 488 (Goat polyclonal) | Invitrogen | Cat#: A11029; RRID:AB_138404 | Immunofluorescence (1:1000) |
| Antibody | Anti-Mouse Alexa Fluor 568 (Goat polyclonal) | Invitrogen | Cat# A11031; RRID:AB_144696 | Immunofluorescence (1:1000) |
| Antibody | Anti-Mouse Alexa Fluor 647 (Goat polyclonal) | Invitrogen | Cat#: A21236; RRID:AB_2535805 | Immunofluorescence (1:1000) |
| Antibody | Anti-Rabbit Alexa Fluor 568 (Goat polyclonal) | Invitrogen | Cat#: A11036; RRID:AB_10563566 | Immunofluorescence (1:1000) |
| Antibody | Anti-Rabbit Alexa Fluor 405 (Goat polyclonal) | Invitrogen | Cat# A31556; RRID:AB_221605 | Immunofluorescence (1:1000) |
| Antibody | Anti-Mouse IgG IRDye 680RD (Goat polyclonal) | Licor | Cat# 926-68070; RRID:AB_10956588 | Western blot (1:20,000) |
| Antibody | Anti-Mouse IgG IRDye 800CW (Goat polyclonal) | Licor | Cat# 926-32210; RRID:AB_621842 | Western blot (1:20,000) |
| Antibody | Anti-Rabbit IgG IRDye 680RD (Goat polyclonal) | Licor | Cat# 926-68071; RRID:AB_10956166 | Western blot (1:20,000) |
| Antibody | Anti-Rabbit IgG IRDye 800CW (Goat polyclonal) | Licor | Cat# 926-32211; RRID:AB_621843 | Western blot (1:20,000) |
| Recombinant DNA reagent | Plasmids used | This paper | N/A | *Supplementary file 1* |
| Sequence-based reagent | HiLITR TEV-protease QPCR primers | This paper | N/A | Materials and methods: 'Model selection' |
| Sequence-based reagent | Random hexamer primer | Invitrogen | Cat# N8080127 | |
| Sequence-based reagent | Individual sgRNA sequences used | This paper | N/A | *Supplementary file 1* |
| Sequence-based reagent | sgRNA libraries derived from hCRISPRi-v2 | *Horlbeck et al., 2016* | RRID:Addgene_83969 | *Supplementary file 2* |
| Peptide, recombinant protein | Fibronectin | Millipore | Cat# FC010 | |
| Peptide, recombinant protein | Bovine serum albumin | Fisher BioReagents | Cat# BP1600 | |
| Peptide, recombinant protein | Aprotinin | Sigma | Cat# A1153 | |

*Continued on next page*

*Continued*

| Reagent type (species) or resource | Designation | Source or reference | Identifiers | Additional information |
|---|---|---|---|---|
| Peptide, recombinant protein | Leupeptin | Roche | Cat# 11017101001 | |
| Peptide, recombinant protein | Endoproteinase LysC | Wako Laboratories | Cat# 12505061 | |
| Peptide, recombinant protein | Sequencing-grade trypsin | Promega | Cat# V5111 | |
| Commercial assay or kit | RNeasy Plus Mini Kit | Qiagen | Cat# 74134 | |
| Commercial assay or kit | QIAamp DNA Blood Maxi Kit | Qiagen | Cat# 51194 | |
| Commercial assay or kit | BCA Assay Kit | Pierce | Cat# 23225 | |
| Commercial assay or kit | MycoAlert Mycoplasma detection kit | Lonza | Cat# LT07-118 | |
| Chemical compound, drug | 1% penicillin-streptomycin | Corning | Cat# 30-002 CI | |
| Chemical compound, drug | GlutaMAX | Gibco | Cat# 35050061 | |
| Chemical compound, drug | Puromycin | Sigma | Cat# P8833 | |
| Chemical compound, drug | Blasticidin | Corning | Cat# 30-100-RB | |
| Chemical compound, drug | Hygromycin | Corning | Cat# 30-240-CR | |
| Chemical compound, drug | Geneticin G418 | Thermo Fisher | Cat# 10131035 | |
| Chemical compound, drug | Polyethyleneimine (PEI) | Polysciences | Cat# 24765-1 | |
| Chemical compound, drug | Polybrene | Millipore | Cat# TR-1003-G | |
| Chemical compound, drug | Doxycycline | Sigma | Cat# C9891 | |
| Chemical compound, drug | MitoTracker Deep Red FM | Invitrogen | Cat# M22426 | |
| Chemical compound, drug | Paraformaldehyde | RICCA | Cat# 3180 | |
| Chemical compound, drug | Triton X-100 | Sigma | Cat# T9284 | |
| Chemical compound, drug | TMTpro isobaric mass tagging reagent | Thermo | Cat# A44520 | |
| Software, algorithm | CasTLE | *Morgens et al., 2016* | https://bitbucket.org/dmorgens/castle/src | |
| Software, algorithm | Bowtie 2 | *Langmead and Salzberg, 2012* | RRID::SCR_016368 | |
| Software, algorithm | SH800S Cell Sorter Software (versions 2.1.2, 2.1.5) | SONY | N/A | |
| Software, algorithm | Everest (version 2.3) | BioRad | N/A | |
| Software, algorithm | FlowJo (version 10.7.1) | FlowJo | N/A | |

*Continued*

| Reagent type (species) or resource | Designation | Source or reference | Identifiers | Additional information |
|---|---|---|---|---|
| Software, algorithm | SlideBook 5.0 software | Intelligent Imaging Innovations | N/A | |
| Software, algorithm | StepOne Software (version 2.2.2) | Applied Biosystems | N/A | |
| Software, algorithm | Limma (version 3.42.2) | *Smyth, 2004* | RRID:SCR_010943 | |
| Other | Fetal bovine serum | Avantor | Cat# 97068-085 | |
| Other | SuperScript III Reverse Transcriptase | Invitrogen | Cat# 18080093 | |
| Other | RiboLock RNAse inhibitor | Thermo Scientific | Cat# EO0382 | |
| Other | Maxima SYBR Green/ROX qPCR Master Mix | Thermo Scientific | Cat# K0221 | |
| Other | Herculase II Fusion | Agilent | Cat# 600679 | |
| Other | Protease Inhibitor Cocktail | Sigma | Cat# P8849 | |
| Other | Precision Plus Protein All Blue Prestained Standards | BioRad | Cat# 1610373 | |

## Mammalian cell culture

HEK293T cells (ATCC) were cultured as a monolayer in a 1:1 DMEM/MEM mixture (Corning 10-017; Corning 15-010) supplemented with 10% fetal bovine serum (FBS; Avantor 97068-085) and 1% penicillin-streptomycin (Corning 30-002CI , final concentration 1 U/mL penicillin and 100 µg/mL streptomycin) at 37 °C with 5 % $CO_2$. Cell line was authenticated by vendor and confirmed free of mycoplasma by PCR test.

K562 cells (ATCC) were cultured in suspension in RPMI 1640 (Corning 15-040) supplemented with 10% FBS (Avantor 97068-085), 1% penicillin-streptomycin (Corning 30-002CI , final concentration 1 U/mL penicillin and 100 µg/mL streptomycin) and 1% GlutaMAX (Gibco 35050061) at 37 °C with 5 % $CO_2$, while subject to 30 rpm linear shaking. For large-scale screens, K562 cells were cultured in spinner flasks (BELLCO 1965-83005) while subject to magnetic stirring at 60 RPM. Cell line was authenticated by vendor and confirmed free of mycoplasma by MycoAlert kit (Lonza LT07-118).

HeLa cells (CVCL_1922) were cultured as a monolayer in Roswell Park Memorial Institute 1640 (Corning 15-040) supplemented with 10% FBS (Avantor 97068-085), 1% penicillin-streptomycin (Corning 30-002CI , final concentration 1 U/mL penicillin and 100 µg/mL streptomycin) and 1% GlutaMAX (Gibco 35050061) at 37 °C with 5% $CO_2$. For fluorescence microscopy experiments, cells were plated on 7 mm × 7 mm glass coverslips in 48-well plates. The coverslips were pretreated with 50 mg/mL fibronectin (Millipore FC010) in identical culturing medium and conditions for at least 4 hr in order to improve cell adherence. Cell line was authenticated by vendor and confirmed free of mycoplasma by MycoAlert kit (Lonza LT07-118).

## Lentivirus generation and stable integration of constructs

Lentivirus was generated by transfection of lentiviral vector (1000 ng) and packaging plasmids pCMV-dR8.91 (900 ng) and pCMV-VSV-G (100 ng) with 12 µL of polyethyleneimine (PEI, 1 mg/mL; Polysciences 24765-1) into HEK293T cells that had been grown to 60–80% confluence in 6-well plates. Total volume of media was 2 mL per transfection. About 48 hr after transfection, the cell medium was harvested in 0.5 mL aliquots and flash-frozen in liquid nitrogen, then stored at –80 °C. Prior to infection, viral aliquots were thawed at 37 °C.

For larger-scale lentivirus generation, lentiviral sgRNA vector libraries (8000 ng) and packaging plasmids (8000 ng total, same weight ratios as above), with 50 µL of PEI, were transfected into HEK293T cells cultured in 15 cm dishes. Total volume of media was 30 mL per transfection. 48 hr after transfection, the cell medium was harvested, and an additional 30 mL of media was added to each plate. 24 hr later, this media was combined with the media from the 48 hr time point and filtered through a 0.45 µm syringe filter (Millipore SLHV033RB; BD 309653). One 15 cm dish was transfected for each 35 million K562 cells to be transduced.

To infect K562 cells, 50,000–250,000 cells in log-phase growth were combined with one or two viral aliquots and 1.6 µL of polybrene (10 mg/mL; Millipore TR-1003-G) in a total volume of 2 mL in a 24-well plate format. The plates were subjected to centrifugation at 1000 × g and 33 °C for 2 hr. For large-scale experiments, infections were performed in 6-well plates, and the volume of reagents and number of plates used was scaled up in proportion to the desired number of cells infected. HeLa cells were infected by adding one or two viral aliquots to the media of a 6-well plate when the cells had grown to 15–30% confluency. For both K562 and Hela Cell, selection was initiated 2 days after infection with 0.5 µg/mL puromycin (Sigma P8833), which was increased in concentration to 1 µg/mL over the next 2 days and supplied for a total of 3–6 days. Some plasmids instead required selection with blasticidin (Corning 30-100-RB, starting with 4 µg/mL and increased to 8 µg/mL over a selection time course of 5–7 days) or with hygromycin (Corning 30-240-CR, starting with 100 µg/mL and increased to 200 µg/mL over a selection time course of 5–7 days), or with geneticin (Thermo Fisher #10131035, starting with 50 µg/mL and increased to 100 µg/mL over a selection time course of 5–7 days).

## Generation of clonal cell lines

To generate clonal K562 cell lines, cell lines were first generated with stable integration and selection of all desired constructs. Cell density was estimated using a Countess II FL automated cell counter, and cells were serially diluted and plated in a 48-well plate format at a target density of 0.2 cells/well. After 1–2 weeks of expansion, clonal cell lines were selected for desired levels of construct expression and HiLITR response, as assessed by flow cytometry analysis.

Clonal HeLa cell lines were generated from cell cultures with stably integrated and selected constructs. After lifting and separating cells with trypsin (Corning 25-053), serial dilutions were plated on 10 cm cell culture dishes. A day after plating, individual clones were identified with an Olympus CKX31 benchtop inverted microscope. After 1–2 weeks of expansion, previously identified colonies were isolated with a cloning cylinder (Millipore TR-1004), lifted with trypsin, and transferred to a 6-well plate for further expansion. Clonal lines were then selected for desired levels of construct expression, as assessed by flow cytometry.

## Immunofluorescence staining and fluorescence microscopy

Cells were incubated with 400 ng/mL doxycycline (Sigma D9891) upon plating if a doxycycline-inducible fluorescent construct was to be Imaged. Roughly 12–16 hr after plating, cells were fixed with 4% paraformaldehyde (RICCA 3180) in phosphate-buffered saline (PBS) for 15 min. For immunofluorescence experiments with K562 cells, the plates containing the cells were subjected to centrifugation at 1000 × g during fixation. After fixation, cells were washed with PBS, then permeabilized with 0.2% triton X-100 (Sigma T9284) in PBS for 10 min. After washing again with PBS, cells were incubated with primary antibody for 1 hr in 2 % BSA (Fisher BioReagents BP1600) in PBS, then washed with PBS and incubated with secondary, fluorophore-conjugated antibody in 2 % BSA in PBS for 30 min, followed by a final wash before imaging. During wash steps, media was removed with vacuum aspiration at the lowest possible pressure setting. On occasions where media removal was performed by hand, additional PBS washes were incorporated between steps.

Imaging was performed with a Zeiss Axio Observer.Z1 microscope with a Yokogawa spinning disk confocal head, Cascade IIL:512 camera, a Quad-band notch dichroic mirror (405/488/568/647 nm), and 405 nm, 491 nm, 561 nm, and 640 nm lasers (all 50 mW). Images were captured through a 63 × or 100× oil-immersion objective for the following fluorophores: BFP and Alexa Fluor 405 (405 laser excitation, 445/40 emission), EGFP and Alexa Fluor 488 (491 laser excitation, 528/38 emission), mCherry and Alexa Fluor 568 (561 laser excitation, 617/73 emission), and Alexa Fluor 647 (647 laser excitation, 700/75 emission). Differential interference contrast (DIC) images were also obtained. Image acquisition times ranged from 50 to 250 ms per channel, and images were captured as the average of two or three such exposures in rapid succession. Image acquisition and processing was carried out with the SlideBook 5.0 software (Intelligent Imaging Innovations, 3i).

Primary antibodies used in imaging include the following: anti-V5 (Mouse, Invitrogen R960); anti-TOMM20 (Rabbit, Abcam ab186735); anti-GRASP65 (Mouse, Santa Cruz sc-374423); anti-CANX (Rabbit, Thermo Fisher PA5-34754); anti-PEX14 (Rabbit, Proteintech 10594-1-AP); anti-RCN2 (Rabbit, Thermo Fisher PA5-56542); anti-AKAP1 (Mouse, Santa Cruz sc-135824); and anti-MAVS (Mouse, Santa Cruz sc-166583).

Secondary antibodies used in imaging include the following: anti-mouse Alexa Fluor 488 (Goat, Invitrogen A11029); anti-mouse Alexa Fluor 568 (Goat, Invitrogen A11031); anti-mouse Alexa Fluor 647 (Goat, Invitrogen A21236); anti-rabbit Alexa Fluor 568 (Goat, Invitrogen A11036); and anti-rabbit Alexa Fluor 405 (Goat, Invitrogen A31556).

MitoTracker Deep Red FM (Invitrogen M22426) was also used for imaging.

## FACS analysis and sorting

FACS analysis and sorting of K562 cells were carried out with a SONY SH800S cell sorter equipped with four collinear excitation lasers (405 nm, 488 nm, 561 nm, and 638 nm; all 30 mW), using a 100 μm sorting chip. The 638 nm laser was disabled during experiments. Additional fluorescent cell cytometry analysis of K562 cells and HeLa cells was performed using a BioRad ZE5 cell analyzer with four parallel excitation lasers (405 nm – 100 mW, 488 nm – 100 mW, 561 nm – 50 mW, and 640 nm – 100 mW). For experiments using the SONY SH800S, instrumental analysis and data processing were performed using the SONY Cell Sorter Software, versions 2.1.2 and 2.1.5. For experiments with the BioRad ZE5, instrumental analysis was performed using Everest software version 2.3 (BioRad) and data processing was performed using FlowJo version 10.7.1.

The following scatter conditions and fluorophores were measured in FACS experiments (note that for the collinear laser configuration of the SONY instrument, all three active lasers engage in simultaneous excitation): forward scatter ('FSC'; SONY: 488/17 emission; BioRad: 488 excitation, 488/10 emission), back/side scatter ('BSC'/'SSC'; SONY: 488/17 emission; BioRad: 488 excitation, 488/10 emission), BFP (SONY: 450/50 emission; BioRad: 405 excitation, 460/22 emission), EGFP (SONY: 525/50 emission; BioRad: 488 excitation, 509/24 emission), mCherry (SONY: 600/60 emission; BioRad: 561 excitation, 615/24 emission). For first experiments with a given laser configuration, appropriate single-fluorophore compensation controls were included. The corresponding compensation matrix was applied to future experiments using the same laser configuration.

A short series of gates was used to focus sorting and analysis on the desired population of cells. Live and dead cells were first separated by plotting back/side scatter area (BSC-A/SSC-A) against forward scatter (FSC-A), and dead or dying cells were excluded by drawing a gate that omitted cells with a high BSC/SSC:FSC ratio, yielding population $P_1$. From $P_1$, single cells were separated from cell doublets by plotting forward scatter height (FSC-H) against width (FSC-W) and drawing a gate around the predominant population with lower FSC-W values, yielding population $P_2$. For experiments featuring sgRNA constructs with BFP expression indicator (particularly the experiments with sgRNA libraries), population $P_2$ was further resolved into population $P_3$ by plotting a histogram of BFP values and collecting only cells with high expression of BFP (and sgRNA by proxy, omitting cells lacking sgRNA or in the bottom 10% of the sgRNA-positive peak of the histogram). Finally, for some experiments in which samples consisted of homogenous cell populations (with identical reporter, TF, TEV protease, and sgRNA constructs integrated), population $P_2$ or $P_3$ was refined to population $P_4$, the population of cells expressing TEV protease (which was fused to EGFP) by plotting FSC-A against EGFP and drawing a gate around the cluster of EGFP-positive cells.

Samples were maintained on ice prior to instrumental analysis. For the large-scale sorting experiments, the acquisition and collection chambers of the SONY SH800S were maintained at 4 °C. Sorted cells were collected in 15 mL conical tubes containing 5 mL of HEPES-buffered RPMI (Sigma R7388) supplemented with 30% FBS (Avantor 97068-085). During sorting, collections tubes that were filled were subjected to centrifugation at 1000 × g, and the media was removed and replaced with HEPES-buffered RPMI with 10% FBS. After the conclusion of all sorting, collected cells were pooled by sample and sort condition, pelleted again by centrifugation and removal of media, and flash-frozen in liquid nitrogen or immediately subjected to sequencing library preparation.

## HiLITR activation

For small-scale analyses, K562 cells were grown in 6-well plate format (at 50,000–500,000 cells/mL), while for large-scale sorting experiments (such as the whole-genome selection), the cells were grown in T125 flasks (at 300,000–600,000 cells/mL). Cells were incubated with 400 ng/mL doxycycline to induce expression of the TEV protease component roughly 16–24 hr prior to light stimulation (experiments where conditions differ noted in the text). Upon addition of doxycycline, the plates or flasks were wrapped completely in tin foil, exposing only the vented flask cap, where applicable. Following

doxycycline incubation, cells were stimulated with 450 nm blue light from a 28.8 × 28.8 cm$^2$, 22 W panel (26.5 mW/cm$^2$; Yescom YES3110) for a period of 2–8 min (depending on the experiment and sample). Plates or flasks were placed directly on top of the panel and agitated by hand once every 1–2 min to prevent settling of cells. Light stimulation was carried out at room temperature in a dark room with only red light sources as additional illumination for visual aid. Following light stimulation, cells were returned to tin foil wrapping and replaced in the 37 °C, 5 % CO$_2$ incubator for 8–16 hr to allow for expression of the reporter construct. After expression, K562 cells were transferred to appropriate tubes for FACS analysis/sorting and placed on ice. For small-scale experiments, cells were sorted in their native RPMI medium. For the large-scale sorts, the cells were collected by centrifugation at 1000 × and the media was removed and replaced with HEPES-buffered RPMI with 10 % FBS. During this medium-replacement step, cells were concentrated to a density of 8–12 million cells/mL.

For HiLITR experiments with HeLa cells, light stimulation was carried out in an identical manner, except cells were cultured on 7 mm × 7 mm glass coverslips, they were not agitated during light stimulation, and they were subjected to immunofluorescence staining and fluorescence microscopy after reporter expression.

## Model selection

A clonal K562 line was generated with a 'matched' HiLITR configuration, bearing a TA, mitochondrial TEV protease component and an SA, mitochondrial TF component. The clone was selected on the bases of good light/dark sensitivity, typical TEV protease expression levels, and HiLITR activation that was not atypically robust. Two additional K562 cell lines with 'mismatched' HiLITR configurations were generated, bearing the same mitochondrial TF and either a TA, ER TEV protease component or an NES-tagged, cytosolic TEV protease component. One day prior to selection, the density of each cell line was estimated using a Countess II FL automated cell counter. For each mismatched-HiLITR cell line, the matched-HiLITR clonal line was mixed at a 1:20, 1:2, 5:1, and 50:1 ratio of matched to mismatched cells, creating a calibration series over four orders of magnitude (~200 thousand cells/sample). Additional 1:20 population mixtures (~2 million cells/sample), as well as unmixed cell lines (~200 thousand cells/sample), were then subjected to the HiLITR activation protocol (doxycycline-induced TEV protease expression, 3.5 min light stimulation). Unmixed cell lines were analyzed by FACS (SONY SH800S), and the resulting activation profiles were used to design gates to maximally enrich the matched-HiLITR cell line from the pooled mixture. Just prior to sorting, a 'pre-sort' baseline of the mixed cell line sample was set aside from each pooled mixture. Sorting was conducted for about 20 min per sample, and about 150,000 cells were collected per sort. Immediately after sorting, RNA was extracted from the post-sort collected, pre-sort baseline, and calibration series populations.

RT-qPCR analysis was performed to measure the levels of matched (mitochondrial) TEV protease transcript and mismatched (cytosolic or ER) TEV protease transcript in each sample. Three technical replicates were measured per sample (separated into replicates after RNA extraction and reverse transcription), with the following sequencing primers:

- Forward primer (same for all transcripts): 5'-CATGGTGGAATTCGGTTCCACG-3'
- Mitochondrial TEV protease reverse primer: 5'-GGTGAGGGCCTTCCACTACC-3'
- ER TEV protease reverse primer: 5'-GGACTCCACGGTGGTGATTC-3'
- Cytosolic TEV protease reverse primer: 5'-CGGCCAGCTCTCCACTACC-3'

A 60 °C annealing temperature was used in the qPCR reaction. For each sample, the ratio of matched protease transcript to mismatched protease transcript was used as a proxy for the ratio of cells from the corresponding populations. Comparison of the transcript ratios from the pre-sort and post-sort samples to the calibration series enabled calculation of the absolute ratio of matched protease to mismatched protease cells in each sample, from which the corresponding enrichment of cells bearing the matched protease could be derived.

## RNA extraction

RNA extraction was performed using a RNeasy Plus Mini Kit (Qiagen 74134). Extraction was performed in accordance with the protocol provided in the kit, at a 350 µL scale for pelleted cells.

## RT-qPCR analysis

To convert RNA to single-stranded cDNA for qPCR analysis, 8 µL of extracted RNA was mixed with 1 µL of 10 mM dNTPs and 1 µL of 50 µM random hexamer primer (Invitrogen N8080127). The sample was heated to 65 °C for 5 min and then stored on ice for 1 min. The sample was then added to a mixture of 1 µL SuperScript III RT, 4 µL 5× X First-strand Buffer, and 2 µL 0.1 µM DTT (all from Invitrogen 18080044), with 1 µL RiboLock RNAse inhibitor (Thermo Scientific EO0382), and 2 µL RNAse-free water. Single-stranded DNA was generated by placing the mixture on a thermocycler for 10 min at 25 °C, followed by 1 hr at 55 °C and 15 min at 40 °C, before holding at 4 °C.

To analyze cDNA by qPCR, cDNA was first diluted 25-fold in water. Subsequently, 2 µL of cDNA was mixed with 2.4 µL water, 0.3 µL each of 10 µM forward and reverse primers, and 5 µL of Maxima SYBR Green/ROX qPCR Master Mix (Thermo Scientific K0221). Samples were prepared on ice and arranged in a MicroAmp 48-well reaction place (Applied Biosystems 4375816), which was sealed with MicroAmp optical adhesive film (Applied Biosystems 4375323). Instrumental analysis was performed on a StepOne Real-Time PCR system (Applied Biosystems 436907) using StepOne Software (version 2.2.2). Samples were quantified over 40 cycles of amplification, followed by melt curve analysis for quality control. Count values for each sample were obtained using automatic thresholding performed by the software, and count values were exported to Microsoft Excel for additional analysis.

## Whole-genome selection

The top five sgRNA per gene from a genome-wide CRISPRi library (*Horlbeck et al., 2016*) were used for the genome-wide CRISPRi screen. After generation of lentivirus, the library was infected into the clonal K562 cell line generated for the model selection (mitochondrial TF, TA mitochondrial TEV protease). Infection was performed with 280 million K562 cells. Based on FACS analysis of BFP-positive cells, multiplicity of infection was 0.4, for a theoretical coverage of 1100× per library element. We selected for sgRNA incorporation with puromycin, split samples into two technical replicates, and 36 hr prior to FACS sorting, induced TEV protease expression with doxycycline. Cells were maintained at or above coverage for the culture duration. The cells were transferred to T150 flasks (60 mL) for light stimulation 12 hr prior to sorting, then returned to the spinner flask for reporter expression. Sorting was performed 9 days after infection. About 200 million cells were analyzed by FACS for each technical replicate. Gates were set to collect cells with the top 15% and bottom 15% of mCherry reporter expression, representing the cells with the greatest and least HiLITR activity. Based on sorting purity parameters, about 18 million cells were collected for each gate. Genomic DNA was harvested from cells (Qiagen 51192) immediately after completion of sorting.

## Matched sublibrary selections

The library for the three matched selections was designed based on the results from the whole-genome selection. The library featured sgRNAs targeting 586 genes (five sgRNAs per gene) as well as 500 nontargeting controls. Genes targeted were selected based on significance in the whole-genome selection and on lack of clear annotation related to transcription or translation. A smaller set of genes corresponding to known protein trafficking pathways was also included.

In addition to the clonal K562 cell line that was previously used in the whole-genome selection (TA cell line), two additional clonal K562 cell lines were generated. The first clonal cell line expressed the same mitochondrial TF as the whole-genome selection cell line, but the TEV protease was an SA mitochondrial construct (SA cell line). The second clonal cell line expressed a mutagenized TA mitochondrial TEV protease, which was sensitized for misincorporation into the ERM, and the TF construct in this cell line was localized to the ERM (ER cell line).

Each cell line was transduced with lentivirus of the sgRNA sublibrary at 50 million cell scale for the TA and SA cell lines, and 70 million cell scale for the ER cell line. Cell lines were split into two biological replicates immediately after infection, before cells began to divide. Multiplicity of infection was estimated as follows, based on proportion of BFP-positive, sgRNA-expressing cells: 0.8 for the TA cell line (5700× coverage per biological replicate), 1.4 for the SA cell line (10,000× coverage), and 0.16 for the ER cell line (1600× coverage). Cells were cultured in T150 flasks with linear shaking, and coverage levels were maintained during cell culturing and selection. TEV protease expression was induced with doxycycline 36 hr prior to sorting, and light stimulation was performed 12 hr prior to sorting. Sorting was performed 11 days after infection. For the TA cell line, cells with the top and bottom 16% of

mCherry reporter expression were collected (11 million cells per condition). For the SA cell line, cells with the top 16% and bottom 33% of mCherry reporter expression were collected (12 million and 24 million cells, respectively). For the ER cell line, cells with the top 9% and bottom 30% of mCherry reporter expression were collected (6 million and 22 million cells, respectively). Genomic DNA was harvested from cells immediately after completion of sorting.

## Sequencing library preparation

The integrated sgRNA library was PCR amplified and separately barcoded for each collected population with Herculase II Fusion DNA Polymerase (Agilent 600677). Samples were then pooled and sequenced on an Illumina NextSeq flow cell with aligned read counts as follows for each screen:

> Whole-genome screen: TR1 (HiLITR active) – 51 million; TR1 (HiLITR inactive) – 51 million; TR2 (HiLITR active) – 49 million; TR2 (HiLITR inactive) – 47 million.
> TA screen: TR1 (HiLITR active) – 6 million; TR1 (HiLITR inactive) – 8 million; TR2 (HiLITR active) – 7 million; TR2 (HiLITR inactive) – 8 million.
> SA screen: TR1 (HiLITR active) – 9 million; TR1 (HiLITR inactive) – 6 million; TR2 (HiLITR active) – 7 million; TR2 (HiLITR inactive) – 9 million.
> ER screen: TR1 (HiLITR active) – 11 million; TR1 (HiLITR inactive) – 13 million; TR2 (HiLITR active) 11 million; TR2 (HiLITR inactive) – 11 million.

## Data analysis of CRISPR screens

CRISPR screens were analyzed using CasTLE (*Morgens et al., 2016*), a maximum likelihood estimator that determines each gene's effect size based on the enrichment of its sgRNAs relative to a null effect model derived from the enrichments of nontargeting control sgRNAs. The significance of each gene's effect size is tested by evaluating it against the distribution of the estimated effect sizes from random permutations drawn from all targeting sgRNA within the library.

More explicitly, the effect size is the $log_2$-transformed maximum likelihood estimate (using a Bayesian framework) of the change in the ratio of high mCherry to low mCherry cells for knockdown of a given gene, relative to the complement of nontargeting controls. The CasTLE score is twice the log-likelihood ratio of estimated effect size (monotonically increasing with decreasing p-value).

Cloning individual sgRNA lentiviral vectors sgRNA vectors were cloned as follows. The U6:sgRNA_ Puro-T2A-BFP lentiviral vector was digested with BlpI and BstXI. Primers corresponding to the sgRNA target (F – 5′-TTG-[*Guide*]- GTTTAAGAGC-3′; R – 5′-TTAGCTCTTAAAC-[*Guide-rev.comp.*]-CAACAAG-3′) were mixed (10 μL of 10 μM oligo each) and annealed by heating to 95 °C for 5 min, followed by cooling to 25 °C at 5 °C/min. Annealed primers were then diluted 1:20 in water and cloned into the digested vector by T4 ligation.

See *Supplementary file 1* for specific *guide* sequences used.

## mTA* protease immunofluorescence quantification

A clonal HeLa cell line was generated, expressing dCas9-KRAB-BFP and the doxycycline-inducible, mutant TA TEV protease fused to EGFP. The clonal line was separately infected with sgRNAs against genes of interest and nontargeting control. After selection of sgRNA-positive cells with puromycin, samples were plated on coverslips on the eighth day after infection, and TEV protease expression was induced with doxycycline. The following day, cells were fixed, permeabilized, and subjected to immunostaining. Primary: mouse anti-GRASP65 (Golgi; Santa Cruz sc-374423, 1:500 dilution) and rabbit anti-TOMM20 (mitochondria; Abcam ab186735, 1:500 dilution); secondary: goat anti-mouse Alexa Fluor 647 (Invitrogen A21236, 1:1000 dilution) and goat anti-rabbit Alexa Fluor 568 (Invitrogen A11036, 1:1000 dilution).

The resulting images were analyzed with SlideBook 5.0 software (Intelligent Imaging Innovations, 3i), as follows. Cells that were completely present in the image and that had clear signal in the fluorescent channels corresponding to GRASP65, TOMM20, and the mutant TA TEV protease were bounded to generate unique image objects. For each object, a mask was created on pixels that exceeded a threshold GRASP65 signal. A second mask was created for pixels that exceeded a threshold TOMM20 signal but not the GRASP65 threshold. Average EGFP intensity was calculated for each mask. After measuring all cells, the ratio of average EGFP intensity between the two masks was determined for each cell, then normalized to the median value for cells bearing the nontargeting control sgRNA.

Significance was measured by comparing sample distributions to the nontargeting control using an independent two-sample t-test with equal variance assumption.

In some figures, images were taken across multiple experiments and data is combined in the final figure. This is noted where applicable. Data was combined only for presentational purposes, and not in a manner that affected measurements or statistical calculations. In all cases, all images corresponding to an individual guide were obtained in the same experiment, and each experiment had a corresponding nontargeting control sample. Prior to combination of the experimental data, data within each experiment was standardized to a mean of 1 and standard deviation of 0.16 for the nontargeting control sample. Significance for samples in combined data figures was calculated with respect only to the data from the nontargeting control sample from the same experiment, not the combined sample.

When analyzing localization of the six mutant TA protease candidates, a Golgi marker was not present. Instead, Pearson's coefficient between the protease channel and the TOMM20 channel was calculated for each cell.

## Endogenous protein immunofluorescence quantification

HeLa cells expressing dCas9-KRAB-BFP were infected with sgRNAs and passaged for 8 days with puromycin selection, before plating on glass coverslips. The following day, cells were fixed, permeabilized, and immunostained. Primary: mouse anti-MAVS (Santa Cruz sc-166583, 1:200 dilution) or mouse anti-AKAP1 (Santa Cruz sc-135824, 1:200 dilution) and rabbit anti-TOMM20 (Abcam ab186735, 1:500 dilution). Secondary: goat anti-mouse Alexa Fluor 488 (Invitrogen A11029, 1:1000 dilution) and goat anti-rabbit Alexa Fluor 568 (Invitrogen A11036, 1:1000 dilution).

The resulting images were analyzed with SlideBook 5.0 software (Intelligent Imaging Innovations, 3i), as follows. Cells that were completely present in the image and that had clear signal in the fluorescent channels corresponding to MAVS and TOMM20 were bounded to generate unique image objects. For each object, a mask was created for pixels that exceeded a threshold intensity for both MAVS and TOMM20, based on intensity in the nucleus. A second mask was created for pixels that exceeded the threshold for MAVS but not TOMM20. After background subtraction (using a noncellular region), the percent of non-mitochondrial MAVS was measured as the total MAVS intensity from the MAVS-only mask divided by the sum total MAVS intensity across both masks. Significance was measured by Wilcoxon rank-sum test due to the presence of outliers and skew with certain sgRNAs. AKAP1-stained cells were analyzed in the same manner as MAVS-stained cells.

## Western blots

HeLa cells expressing dCas9-KRAB-BFP were infected with sgRNAs and passaged for 9 days in T25 or T75 flasks with puromycin selection. To harvest, cells were washed twice with DBPS. In 2 mL of DPBS, the cells were dislodged with a cell scraper (Thermo Fisher 179693) and pelleted by centrifugation (500 × g for 3 min). The cell pellets were then resuspended in 1 mL DPBS and pelleted again by centrifugation in 1.5 mL Eppendorf tubes (500 × g for 3 min). Supernatant was removed by aspiration, and the pellets were flash frozen with liquid nitrogen and stored at –80 °C. Later, the pellets were lysed by resuspending in RIPA buffer (50 mM Tris pH 8, 150 mM NaCl, 0.1% SDS, 0.5% sodium deoxycholate, 1% Triton X-100; Sigma T9284) in the presence of 1× protease inhibitor cocktail (Sigma-Aldrich P8849) and 1 mM PMSF. The Eppendorf tubes were incubated for 15 min at 4 °C and vortexed every 3 min for proper sample digestion. Lysates were clarified by centrifugation at 10,000 RPM for 15 min at 4 °C. Protein loading buffer (6×, 20 µL) was mixed with 100 µL of the clarified lysate and boiled for 3 min prior to PAGE gel separation.

Proteins were separated on 9 or 12% SDS-PAGE gels in Tris-Glycine buffer and then were transferred into PVDF membrane (Sigma 05317). The blots were then blocked in 3% BSA (w/v) in TBS-T (Tris-buffered saline, 0.1 % Tween 20) for 45 min at room temperature. Blots were then incubated with primary antibody in 3 % BSA (w/v) in TBS-T for 1 hour at room temperature, washed two times with TBS-T for 10 min each, then stained with secondary antibody in 3% BSA (w/v) in TBS-T for 45 min at room temperature. The blots were washed four times with TBS-T for 5 min each time before imaging on Licor Odyssey CLx imaging system. Quantitation was performed using the software provided by Licor.

Primary

- SAE1 knockdowns: Mouse anti-GAPDH (Santa Cruz sc-32233) – 1:3000; Rabbit anti-SAE1 (Sigma SAB4500028) – 1:500; Rabbit anti-COX4 (Abcam ab16056) – 1:1000; Mouse anti-VDAC1 (Abcam ab14734) – 1:500; Mouse anti-AKAP1 (Santa Cruz sc-135824) – 1:500; Mouse anti-MAVS (Santa Cruz sc-166583) – 1:250; Rabbit anti-SYNJ2BP (Sigma HPA000866) – 1:500; Rabbit anti-FIS1 (Thermo Fisher 10956-1-AP) – 1:1000.
- EMC knockdowns: Mouse anti-GAPDH (Santa Cruz sc-32233) – 1:4500; Rabbit anti-CALR (Thermo Fisher PA3900) – 1:500; Rabbit anti-VTI1B (Abcam ab184170) – 1:250; Rabbit anti-SQS (Abcam ab195046) – 1:500.

Secondary

- SAE1 knockdowns: Goat anti-Mouse IgG IRDye 680RD Polyclonal Antibody (Licor 926-68070) and Goat anti-Rabbit IgG IRDye 800CW Polyclonal Antibody (Licor 926-32211) – 1:20,000.
- EMC knockdowns: Goat anti-Mouse IgG IRDye 800CW Polyclonal Antibody (Licor 926-32210) and Goat anti-Rabbit IgG IRDye 680RD Polyclonal Antibody (Licor 926-68071) – 1:20,000.

## Proteomic profiling

### In-solution digestion

HeLa cell pellets were lysed in-solution with 8 M urea, 75 mM NaCl, 50 mM Tris-HCl pH 8.0, 1 mM EDTA, 2 µg/mL aprotinin (Sigma A1153), 10 µg/mL leupeptin (Roche 11017101001), and 1 mM phenylmethylsulfonyl fluoride (PMSF; Sigma). Protein concentration of cleared lysate was estimated with a bicinchoninic acid (BCA) assay (Pierce 23225). Protein disulfide bonds were reduced with 5 mM dithiothreitol (DTT) at room temperature for 1 hr, and free thiols were alkylated in the dark with 10 mM iodoacetamide (IAM) at room temperature for 45 min. The urea concentration in all samples was reduced to 2 M by addition of 50 mM Tris-HCl, pH 8.0. Denatured proteins were then enzymatically digested into peptides upon incubation first with endoproteinase LysC (Wako Laboratories 12505061) at 25 °C shaking for 2 hr and then with sequencing-grade trypsin (Promega V5111) at 25 °C shaking overnight, both added at a 1:50 enzyme:substrate ratio. Digestion was quenched via acidification to 1% formic acid (FA). Precipitated urea and undigested proteins were cleared via centrifugation, and samples were desalted using 50 mg tC18 1cc SepPak desalt cartridges (Waters 186000308). Cartridges were conditioned with 100% acetonitrile (MeCN), 50% MeCN/0.1 % FA, and 0.1% trifluoroacetic acid (TFA). Samples were loaded onto the cartridges and desalted with 0.1 % TFA and 1 % FA, and were then eluted with 50 % MeCN/0.1 % FA. Eluted samples were frozen and dried via vacuum centrifugation.

### TMT labeling of peptides

Desalted peptides were reconstituted in 30 % MeCN/0.1 % FA and the peptide concentration was quantified with a BCA assay. With 100 µg peptide input per channel, samples were labeled with a TMTpro isobaric mass tagging reagent (Thermo A44520) as previously described (*Zecha et al., 2019*). Samples were reconstituted in 50 mM HEPES, pH 8.5, at a peptide concentration of 5 mg/mL. Dried TMT reagent was reconstituted in 100% anhydrous MeCN at a concentration of 40 µg/µL, added to each sample at a 2:1 TMT:peptide ratio, and allowed to react for 1 hr at 25 °C. Labeling was quenched upon addition of 5 % hydroxylamine to a final concentration of 0.25%, incubating for 15 min at 25 °C. TMT-labeled samples were combined, frozen, and dried via vacuum centrifugation. This dried sample was reconstituted in 0.1 % FA and desalted using a 100 mg tC18 1cc SepPak cartridge as described above. The eluted sample was frozen and dried via vacuum centrifugation.

### Basic reverse phase (bRP) fractionation

Labeled and combined peptides for proteome analysis were fractionated using offline basic reverse-phase (bRP) fractionation as previously described (*Mertins et al., 2018*). The sample was reconstituted in 900 µL bRP solvent A (2% vol/vol MeCN, 5 mM ammonium formate, pH 10.0) and loaded at a flow rate of 1 mL/min onto a custom Zorbax 300 Extend C18 column (4.6 × 250 mm, 3.5 µm, Agilent 770995) on an Agilent 1100 high-pressure liquid chromatography (HPLC) system. Chromatographic separation proceeded at a flow rate of 1 mL/min with a 96 min gradient, starting with an increase to

16% bRP solvent B (90% vol/vol MeCN, 5 mM ammonium formate, pH 10.0), followed by a linear 60 min gradient to 40% that ramped up to 44% and concluded at 60% bRP solvent B. Fractions were collected in a Whatman 2 mL 96-well plate (GE Healthcare) using a horizontal snaking pattern and were concatenated into 24 final fractions for proteomic analysis. Fractions were frozen and dried via vacuum centrifugation.

## Liquid chromatography and mass spectrometry

Sample analysis was performed via coupled nanoflow liquid chromatography and tandem mass spectrometry (LC-MS/MS). Fractions were reconstituted in 3% MeCN/0.1% FA at a peptide concentration of 1 μg/μL. From each fraction, 1 μg sample was loaded for online separation onto an ~25 cm analytical capillary column (360 μm O.D. × 75 μm I.D.), heated to 50 °C and packed with Repro-Sil-Pur C18-AQ 1.9 μm beads (Dr. Maisch GmbH), with a 10 μm electrospray emitter tip. Nanoflow liquid chromatography was performed with an Easy-nLC 1200 system (Thermo), employing a 110 min gradient with varying ratios of solvent A (3% MeCN/0.1% FA) and solvent B (90% MeCN/0.1% FA). Described as min:% solvent B, the steps in the gradient include 0:2, 1:6, 85:30, 94:60, 95:90, 100:90, and 110:50, beginning at a flow rate of 200 nL/min for the first six steps and increasing to 500 nL/min for the final two.

Tandem MS analysis was performed on a Q-Exactive HF-X series mass spectrometer (Thermo). Acquisition was done in data-dependent MS2 mode, picking the top 20 most abundant precursor peaks in an MS1 scan for fragmentation. MS1 scans were collected at a resolution of 60,000, with an automatic gain control (AGC) target of $3 \times 10^6$ ions, or a maximum inject time of 50 ms. HCD-MS2 scans were collected at a resolution of 45,000, with an AGC target of $5 \times 10^4$, or a maximum inject time of 105 ms. The MS2 isolation window was 0.7 m/z, and a collision energy of 29 was used. Ions with a charge state other than 2–6 were excluded, peptide matching was set to 'preferred,' and dynamic exclusion time was set to 15 s. Raw mass spectrometry data is publicly available in MassIVE.

## Data analysis

Mass spectrometry data was processed using Spectrum Mill (Rev BI.07.04.210, proteomics.broadinstitute.org). Extraction of raw files retained spectra within a precursor mass range of 750–6000 Da and a minimum MS1 signal-to-noise ratio of 25. MS1 spectra within a retention time range of ±60 s or within a precursor m/z tolerance of ±1.4 m/z were merged. MS/MS searching was performed against a human UniProt database. Digestion parameters were set to 'trypsin allow P' with an allowance of four missed cleavages. The MS/MS search included fixed modifications, carbamidomethylation on cysteine and TMT on the N-terminus and internal lysine, and variable modifications, acetylation of the protein N-terminus and oxidation of methionine. Restrictions for matching included a minimum matched peak intensity of 30% and a precursor and product mass tolerance of ±20 ppm. Peptide matches were validated using a maximum FDR threshold of 1.2%, limiting the precursor charge range to 2–6. Protein matches were additionally validated, requiring a minimum protein score of 0. Validated data was summarized into a protein-centric table and filtered for fully quantified hits represented by two or more unique peptides. Non-human contaminants and human keratins were removed.

## Statistical approach

Each protein ID was associated with a $\log_2$-transformed expression ratio for every sample condition over the median of all sample conditions. After normalization, a two-sample moderated t-test was performed on the data to compare treatment groups using an internal R-Shiny package based in the limma library. p-Values associated with every protein were adjusted using the Benjamini–Hochberg FDR (*Benjamini and Hochberg, 1995*).

## Resource availability

Lead contact: Further information and requests for resources or reagents should be directed to the lead contact, Alice Ting (ayting@stanford.edu).

## Materials availability

Plasmids generated in the study have been deposited to Addgene or are available upon request (*Supplementary file 1*).

## Acknowledgements

We thank Tina Kim and Kelvin Cho for comments and feedback during manuscript preparation, Wenjing Wang for technical guidance, Shuo Han for advice on clonal cell line generation, and Kaitlyn Spees for assistance with cloning sgRNA library cloning. We are grateful to Chan Zuckerberg Biohub Stanford for access to FACS sorters. This work was supported by NIH R01 MH119353 (AYT), NIH Director's New Innovator Award 1DP2HD084069-01 (MCB), NSF GRFP DGE 1656518 (DY), the Stanford Bio-X Graduate Fellowship Program (RC), the NIST JIMB training program (RC), NIH 2T32HG000044 (RC and DY), a Stanford Center for Systems Biology seed grant (RC and DY), and an EMBO long-term postdoctoral fellowship ALTF 1022-2015 (MIS).

## Additional information

### Funding

| Funder | Grant reference number | Author |
| --- | --- | --- |
| National Institute of Mental Health | MH119353 | Alice Y Ting |
| NIH Office of the Director | 1DP2HD084069-01 | Michael C Bassik |
| National Science Foundation | 1656518 | David Yao |
| Stanford Bio-X | | Robert W Coukos |
| National Institute of Standards and Technology | | Robert W Coukos |
| National Human Genome Research Institute | 2T32HG000044 | Robert W Coukos David Yao |

The funders had no role in study design, data collection and interpretation, or the decision to submit the work for publication.

### Author contributions

Robert Coukos, David Yao, Conceptualization, Funding acquisition, Investigation, Methodology, Validation, Visualization, Writing – original draft, Writing – review and editing; Mateo I Sanchez, Investigation, Methodology, Validation, Writing – review and editing; Eric T Strand, Investigation, Methodology, Writing – review and editing; Meagan E Olive, Data curation, Formal analysis, Investigation, Methodology, Visualization, Writing – review and editing; Namrata D Udeshi, Data curation, Investigation, Methodology, Project administration, Supervision, Writing – review and editing; Jonathan S Weissman, Methodology, Writing – review and editing; Steven A Carr, Data curation, Funding acquisition, Project administration, Resources, Software, Supervision, Writing – review and editing; Michael C Bassik, Conceptualization, Funding acquisition, Methodology, Project administration, Resources, Supervision, Writing – original draft, Writing – review and editing; Alice Y Ting, Conceptualization, Funding acquisition, Methodology, Project administration, Resources, Supervision, Visualization, Writing – original draft, Writing – review and editing

### Author ORCIDs

Robert Coukos http://orcid.org/0000-0002-7307-8293
Mateo I Sanchez http://orcid.org/0000-0003-1359-6969
Eric T Strand http://orcid.org/0000-0002-1488-1043
Michael C Bassik http://orcid.org/0000-0001-5185-8427
Alice Y Ting http://orcid.org/0000-0002-8277-5226

### Decision letter and Author response

Decision letter https://doi.org/10.7554/eLife.69142.sa1
Author response https://doi.org/10.7554/eLife.69142.sa2

## Additional files

### Supplementary files

• Supplementary file 1. Plasmids used in the study and individual sgRNA sequences used. *Plasmids_ Used*: plasmid table for this study. *sgRNAs_Used*: sgRNA sequences used for individual sgRNA sequences.

• Supplementary file 2. Information about sgRNA libraries related to *Figure 2—figure supplement 1* and to *Figure 3*. Sequencing and CasTLE analysis results from the whole-genome screen and sublibrary screens. Comparison of individual validation data to sublibrary screen data. *WGS_ sgRNAs*: sgRNA sequences and target genes in the whole-genome screen. *TA_WGS*: CasTLE analysis of the whole-genome screen (TA screen HiLITR configuration). *Sublibrary_sgRNAs*: sgRNA sequences and target genes in the sublibrary screens. *TA/SA/ER_Sublibrary*: CasTLE analysis of the sublibrary screens (TA/SA/ER screen HiLITR configurations). *Sublibrary_Comparison*: comparison of combined-replicate CasTLE analysis across the TA/SA/ER sublibrary screens. *Hits&Validation*: sublibrary screen data for genes mentioned in main and supplementary figures, with independent validation data appended where applicable.

• Supplementary file 3. Data from the proteomic analysis related to *Figure 5* and *Figure 5—figure supplements 3–5*. *Results_MedNormed*: abundance values and statistical analysis of experimental replicates, normalized to median abundance value in the replicate/column. *Results_MitoNormed*: abundance values and statistical analysis of experimental replicates, normalized to mean abundance value across mitochondrial proteins in the replicate/column. *Results_ER-Normed*: abundance values and statistical analysis of experimental replicates, normalized to mean abundance value across ER proteins in the replicate/column.

• Transparent reporting form

### Data availability

Lead contact: Further information and requests for resources or reagents should be directed to the lead contact, Alice Ting (ayting@stanford.edu) Materials availability: Plasmids generated in the study have been deposited to Addgene or are available upon request (Supplementary file 1) Data and code availability: HiLITR screen sequencing data has been deposited to Dryad (https://doi.org/10.5061/ dryad.tb2rbp00n). The original mass spectra and the protein sequence database used for searches have been deposited in the public proteomics repository MassIVE (http://massive.ucsd.edu) under the accession number MSV000087769 and are accessible at ftp://massive.ucsd.edu/MSV000087769/.

The following dataset was generated:

| Author(s) | Year | Dataset title | Dataset URL | Database and Identifier |
|---|---|---|---|---|
| Yao D | 2021 | HiLITR CRISPR screens | https://doi.org/ 10.5061/dryad. tb2rbp00n | Dryad Digital Repository, 10.5061/dryad.tb2rbp00n |
| Coukos R, Yao D, Sanchez M, Strand E, Olive M, Udeshi N, Weissman J, Carr S, Bassik M, Ting A | 2021 | MassIVE | https://doi.org/10. 25345/C5HZ63 | MassIVE, 10.25345/C5HZ63 |

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
