## [Decision Letter]

**Acceptance summary:**

This is a beautiful study that develops a robust, high-throughput and genome-wide strategy to identify genes that influence protein localization in eukaryotic cells. This new tool will shed lights on the molecular mechanisms of protein trafficking and localization.

**Decision letter after peer review:**

Thank you for submitting your article "An engineered transcriptional reporter of protein localization identifies regulators of mitochondrial and ER membrane protein trafficking in high-throughput screens" for consideration by *eLife*. Your article has been reviewed by 3 peer reviewers, including Heedeok Hong and the Reviewing Editor and Reviewer #1, and the evaluation has been overseen by David Ron as the Senior Editor. The following individuals involved in review of your submission have agreed to reveal their identity: Jonathan P Schlebach (Reviewer #2); Liang Ge (Reviewer #3).

Reviewers agreed that this is an outstanding study and additional experiments are not necessary. It is recommended that authors address the following Essential Revision points and other scientific concerns/suggestions raised by reviewers to further improve the vision, clarity and presentation of this work.

Essential revisions:

1) More in-depth discussion on the limitation and future applicability of this method:

1– (i) While these screens are capable of identifying the central machinery involved in the targeting pathways of TA proteins, chimeric substrates represent an artificial substrate, and the characterization of such substrates may have limited impact on our understanding of topogenic pathways within the cell.

1– (ii) On the perspectives of HiLITR: The current version focuses on single membrane-spanning peptides as a localization signal and it is still unclear how this method can be used to study more complex problems in protein localization (e.g., membrane proteins with multiple TM segments or larger water-soluble domains). In such case, how could the accessibility issue between TF and protease be overcome?

2) More in-depth discussion on the role of the identified genes (SAE1 and EMC10):

Although this manuscript majorly focuses on the tool development, more molecular level explanation on the mechanistic role of SAE1 and EMC10 seems needed. For example, how can EMC10 play an antagonistic role in the insertion of TA proteins in ER membranes and what is its biological implications?

3) More discussion on how the HiLITR activity can be scaled vs the actual contribution of the identified genes:

Signal amplification can be a double-edged dagger since it can magnify small differences more than what is actual. A statement is needed how the HiLITR results can be translated into the actual effect of an identified component (e.g., HILITR vs Western blotting).

4) Call Figure 5D/E in the last paragraph in page 14 (section on EMC2/8/10 knockdown in HeLa).

5) In Figure S4G, could the authors explain why the mito-protease generated more mCherry than the peroxisome-protease since the TF is located on the peroxisome?

*Reviewer #1 (Recommendations for the authors):*

I have one major scientific concern. A major scientific concern:

1. The validity of the antagonizing role of EMC10:

In the HiLITR ER screen of mTA* (Figure 5A), sgRNA-EMC10 yields a large increase in HiLITR signal. However, with the same sgRNA, the Western blotting test for the bona fide EMC client SQS displays a moderate increase (by ~20%, Figure 5E/F). Is this discrepancy due to the difference between mTA* and SQS (i.e., one client is more affected by EMC10 than the other) or to the sensitivity difference between the two methods?

Furthermore, in Figure S15, the band intensity of non-EMC client VTI1B (Western blotting result, Figure S15E) increases with sgRNA-EMC4 (assigned "non-significant" in Figure 5F). The degree of increase is apparently similar to that of the EMC client SQS with sgRNA-EMC10 (Figure S15F, assigned "significant" in Figure 5F). These arguments raise a concern whether the antagonizing role of EMC10 is substantial or minor.

More discussion seems to be needed whether the identified gene products would have uniform or different effects on different clients. Also, it would be nice to discuss how the signal amplification in HiLITR can be interpreted with regards to the actual contribution of the gene products to the TA protein insertion.

*Reviewer #2 (Recommendations for the authors):*

Generally speaking, this article is very well written, and I have few complaints about the technical aspects of these experiments and the associated data. In fact, I believe strongly that this paper goes above and beyond what most would expect. Nevertheless, there were a few places where the text was not so clear, and I suggest a few stylistic changes in order to improve the readability of the paper as follows:

– I found myself referring back to Figures 2 C and D several times in order to make sure I understood how each variation of the screen worked. I think part of the reason this was not immediately clear is that these graphics show you how the system is normally oriented, but does not show how perturbations will effect the system. This is sort of made clear for the untargeted protease controls in Figure 2A, but not for the actual screen itself. I would suggest you make a version of Figure 2C/D where you show the configuration of the screen, then show what a positive hit does to the localization of the sensor and protease in each case (like in 2A). You could split off 2E into another figure, and simply dedicate Figure 2 to these schematics as well as the flow chart. Having a master figure like this would make it much easier for the reader to refer back to what the change in the signal for each screen/ counter screen would indicate at the molecular level.

– In the last paragraph in page 14 (section on EMC2/8/10 knockdown in HeLa) there it is unclear where these data are located. Please indicate the figure number.

– In Figure 5A, the presentation of the three-dimensional scatter is redundant and distracting considering how focused the Figure/ Section is intended to be. I might suggest adding, instead, a table that is focused on hits of interest that just shows their values for each of the three screens. This might make it easy to compare the relevant values across the EMC subunits (and other hits of interest).

– The box and whisker plots displaying the microscopy results have confusing axis labels. If this is indeed the best metric and best title for the axis, I would suggest the authors include a brief explanation of what this metric specifically reflects in quantitative terms within the Results section.

– It is unclear why the authors measure co-localization with a Golgi marker relative to co-localization with mitochondrial markers when validating these hits (i.e. Figure 4). The other parts of the paper suggest the mislocalized protein ends up in the ER. Why not measure co-localization with an ER marker? I am sure the authors chose this marker for a reason. They should at least add one sentence in the Results section explaining this experimental design.

– It is not clear to me why the SAE1 knockdown leads to an increase in the HiLITR signal in the SA assay (Figure 4A). Is there a clear interpretation? Does this matter? A brief explanation of the interpretation of these results in the text is warranted. Even if it is not important, it is a good example to explain how each result is interpreted (SA vs TA vs ER screens). Going through each result one by one might help clarify the logic for interpreting counter screen results. The bottom line is, if the reader can't exactly follow this logic, it could undermine their appreciation for this (admittedly beautiful) experimental design.

*Reviewer #3 (Recommendations for the authors):*

It would be good to have the study quickly published to guide people in the community who are going to develop screening approaches for their interested directions.

Two suggestions:

1. In Figure 2A, the authors proposed three possibilities of the effect of the sgRNA: 1. blocking TA targeting of the protease, 2. affecting the level of the protease, and 3. no effect. I would suggest adding the fourth possibility. It is also likely that the sgRNA may also affect the correct targeting of the membrane anchored TF. To my understanding, as long as the protease is not able to meet the TF, no mCherry could be produced. Again this possiblity could be controlled by the SA experiments the authors have performed.

2. In Figure S4G, could the authors explain why the mito protease generated more mCherry than the peroxisome protease，since the TF is located on the peroxisome?

---

## [Author Response]

Essential revisions:(1) More in-depth discussion on the limitation and future applicability of this method:1– (i) While these screens are capable of identifying the central machinery involved in the targeting pathways of TA proteins, chimeric substrates represent an artificial substrate, and the characterization of such substrates may have limited impact on our understanding of topogenic pathways within the cell.

We have added this caveat when describing the limitations of our approach in the Discussion section. We agree that it is essential to perform follow-up experiments on endogenous protein substrates, which is what we did with SAE1.

1– (ii) On the perspectives of HiLITR: The current version focuses on single membrane-spanning peptides as a localization signal and it is still unclear how this method can be used to study more complex problems in protein localization (e.g., membrane proteins with multiple TM segments or larger water-soluble domains). In such case, how could the accessibility issue between TF and protease be overcome?

HiLITR is modular in its design, so the protease could be fused to any type of protein, including multipass, peripheral, and cytoplasmic proteins. However, the protease must be cytosol-facing under some condition, so that it may release the TF. Some membrane proteins may not have a suitable cytosol-facing fusion site, although the protease is able to tolerate N- and C-terminal tagging and potentially even internal fusion. These points are now included in the edited discussion.

2) More in-depth discussion on the role of the identified genes (SAE1 and EMC10):Although this manuscript majorly focuses on the tool development, more molecular level explanation on the mechanistic role of SAE1 and EMC10 seems needed. For example, how can EMC10 play an antagonistic role in the insertion of TA proteins in ER membranes and what is its biological implications?

We have provided additional content in the Discussion section speculating on the mechanistic effects of SAE1 and EMC10:

– SAE1: “it is possible that SUMOylation of these or other chaperones alters their activity, client specificity, or subcellular distribution. Alternatively, the effects of SAE1 knockdown may be mediated through changes to cell proliferation. SAE1 was one of several hits involved in regulation of the cytoskeleton and of mitosis. In our proteomics experiment, knockdown of SAE1 globally upregulated most mitochondrial proteins (relative to non-mitochondrial proteins), with the exception of tail-anchored proteins. Perhaps cellular and mitochondrial proliferation become uncoupled upon SAE1 knockdown. Tail-anchored proteins, which must be post-translationally inserted into the outer mitochondrial membrane, may be less influenced by the factors which alter mitochondrial proliferation relative to cellular proliferation.”

– EMC10: “We propose that EMC10 plays a role in this regulation. EMC10 sits below the insertase cavity on the lumenal side and engages with both EMC1/7. Knockdown of EMC10 in our experiments produces results concordant with mutation of the EMC1/7 interface (Miller-Vedam et al., 2020). We therefore speculate that EMC10 stabilizes the EMC1/7 junction and the closed insertase conformation, and that EMC10 may dissociate from the EMC to promote increased insertase activity.”

3) More discussion on how the HiLITR activity can be scaled vs the actual contribution of the identified genes:Signal amplification can be a double-edged dagger since it can magnify small differences more than what is actual. A statement is needed how the HiLITR results can be translated into the actual effect of an identified component (e.g., HILITR vs Western blotting).

In general, HiLITR seems to be more sensitive than direct measures of endogenous protein levels, which can be observed in the Western blotting and proteomics data related to SAE1 knockdown and in the Western blotting related to EMC10 knockdown. This is likely a function of the signal amplification of HiLITR, as the reviewer notes, and the use of clonal selection for highly sensitive cell lines. In theory, if there are perturbations that will be known to give specific effect sizes, they could be used to calibrate the HiLITR readout. Otherwise, we would recommend against imputing a specific effect size from HiLITR results. We have added these comments to the discussion.

4) Call Figure 5D/E in the last paragraph in page 14 (section on EMC2/8/10 knockdown in HeLa).

Done.

5) In Figure S4G, could the authors explain why the mito-protease generated more mCherry than the peroxisome-protease since the TF is located on the peroxisome?

We believe this was a combination of two factors. First, despite several attempts, we were unable to generate a peroxisomal protease that localizes exclusively to peroxisomes. Imaging of the protease in Figure S4F suggests it is also localized to some extent to the mitochondria and/or ER, although this was not directly confirmed. Second, HiLITR activation increases with greater protease expression, and the mitochondrial protease is expressed at higher levels than the peroxisomal protease (compare 1st and 2nd FACS plots in row 2 of Figure 1 —figure supplement 4G). We have added this comment to the text accompanying figure 1 —figure supplement 4.

Reviewer #1 (Recommendations for the authors):I have one major scientific concern.1. The validity of the antagonizing role of EMC10:In the HiLITR ER screen of mTA* (Figure 5A), sgRNA-EMC10 yields a large increase in HiLITR signal. However, with the same sgRNA, the Western blotting test for the bona fide EMC client SQS displays a moderate increase (by ~20%, Figure 5E/F). Is this discrepancy due to the difference between mTA* and SQS (i.e., one client is more affected by EMC10 than the other) or to the sensitivity difference between the two methods?

We have a few possible explanations for the magnitude difference seen in our ER-screen versus the SQS western blots:

– As discussed above, HiLITR is more sensitive than WB due to signal amplification.

– In addition, the HiLITR assay with the mTA* protease was performed in a clonal cell line, which was selected precisely for its sensitivity to perturbation.

– There are also biophysical differences between the mTA* protease reporter and SQS. The mTA* transmembrane domain is a hybrid substrate (combining features of mito TA and ER TA proteins) that is not optimized for ER insertion. Consequently, when we knock down EMC10, increased permissiveness at the ER membrane might strongly impact the population of mTA* protease which would be otherwise targeted to the mitochondria. In contrast, SQS is already an ideal substrate of the EMC. Increased permissiveness may salvage some SQS that would otherwise fail to insert and be degraded, but most SQS is already inserted into the ER membrane and the room for improvement is modest.

Furthermore, in Figure S15, the band intensity of non-EMC client VTI1B (Western blotting result, Figure S15E) increases with sgRNA-EMC4 (assigned "non-significant" in Figure 5F). The degree of increase is apparently similar to that of the EMC client SQS with sgRNA-EMC10 (Figure S15F, assigned "significant" in Figure 5F). These arguments raise a concern whether the antagonizing role of EMC10 is substantial or minor.

In Figure 5F (now 6F), one replicate for VTI1B showed greater intensity from the EMC4 knockout sample. However, the other two replicates showed levels that were very similar to non-targeting control. For the SQS staining, we observed consistent and similar increases in protein levels in all three EMC10 knockout replicates. Since the insertase activity of EMC is constitutive, the antagonism by EMC10 is probably somewhat minor. We speculate that EMC10 can adopt different conformations within the EMC—or even dissociates completely—in order to boost insertase activity at critical junctures.

More discussion seems to be needed whether the identified gene products would have uniform or different effects on different clients. Also, it would be nice to discuss how the signal amplification in HiLITR can be interpreted with regards to the actual contribution of the gene products to the TA protein insertion.

Thank you. We have responded to these points above and made corresponding edits to the manuscript to clarify these points.

Reviewer #2 (Recommendations for the authors):Generally speaking, this article is very well written, and I have few complaints about the technical aspects of these experiments and the associated data. In fact, I believe strongly that this paper goes above and beyond what most would expect. Nevertheless, there were a few places where the text was not so clear, and I suggest a few stylistic changes in order to improve the readability of the paper as follows:– I found myself referring back to Figures 2 C and D several times in order to make sure I understood how each variation of the screen worked. I think part of the reason this was not immediately clear is that these graphics show you how the system is normally oriented, but does not show how perturbations will effect the system. This is sort of made clear for the untargeted protease controls in Figure 2A, but not for the actual screen itself. I would suggest you make a version of Figure 2C/D where you show the configuration of the screen, then show what a positive hit does to the localization of the sensor and protease in each case (like in 2A). You could split off 2E into another figure, and simply dedicate Figure 2 to these schematics as well as the flow chart. Having a master figure like this would make it much easier for the reader to refer back to what the change in the signal for each screen/ counter screen would indicate at the molecular level.

This is an excellent suggestion. We have modified the figure to show how perturbations will affect the readout for each screen.

– In the last paragraph in page 14 (section on EMC2/8/10 knockdown in HeLa) there it is unclear where these data are located. Please indicate the figure number.

Noted and fixed

– In Figure 5A, the presentation of the three-dimensional scatter is redundant and distracting considering how focused the Figure/ Section is intended to be. I might suggest adding, instead, a table that is focused on hits of interest that just shows their values for each of the three screens. This might make it easy to compare the relevant values across the EMC subunits (and other hits of interest).

We believe the scatter plot adds value because it clearly shows that EMC members have among the most pronounced effects within the ER screen, and it also demonstrates that the EMC (and EMC10 in particular) is in a distinct class of its own with respect to other genes targeted in the screens. However, in order to provide readers with a quick summary of results from the EMC, we have added the suggested table to the figure.

– The box and whisker plots displaying the microscopy results have confusing axis labels. If this is indeed the best metric and best title for the axis, I would suggest the authors include a brief explanation of what this metric specifically reflects in quantitative terms within the Results section.

We have added some language to the text and figure 4 legend to clarify the metric, as follows:

– Results: “…knockdown of SAE1 increases the mislocalization of this construct to ER/Golgi compartments, as measured by the ratio of GFP overlapping with Golgi versus mitochondrial markers.

– Legend: “The value plotted is the mean intensity of GFP-protease signal colocalized with Golgi divided by mean signal colocalized with mitochondria.”

A comprehensive description of how measurements were acquired and quantified is provided in the methods.

– It is unclear why the authors measure co-localization with a Golgi marker relative to co-localization with mitochondrial markers when validating these hits (i.e. Figure 4). The other parts of the paper suggest the mislocalized protein ends up in the ER. Why not measure co-localization with an ER marker? I am sure the authors chose this marker for a reason. They should at least add one sentence in the Results section explaining this experimental design.

This is a good question, and we have added some clarifying language in the description of tail-anchored protein trafficking, the gain-of-signal ER screen, and validation of hits. To address the question directly here, the literature on tail-anchored protein trafficking has established that all TA proteins of the endomembrane system are first inserted at the ER. The mTA* protease must also be mistargeted to the ER, but upon further anterograde trafficking through the endomembrane it accumulates most heavily in the Golgi. For this reason and for other technical reasons, it was easiest to image the Golgi reservoir of mTA* protease as an indicator of mislocalization.

– It is not clear to me why the SAE1 knockdown leads to an increase in the HiLITR signal in the SA assay (Figure 4A). Is there a clear interpretation? Does this matter? A brief explanation of the interpretation of these results in the text is warranted. Even if it is not important, it is a good example to explain how each result is interpreted (SA vs TA vs ER screens). Going through each result one by one might help clarify the logic for interpreting counter screen results. The bottom line is, if the reader can't exactly follow this logic, it could undermine their appreciation for this (admittedly beautiful) experimental design.

This is a great question, and something we have been trying to determine ourselves. The proteomics data which we acquired suggests that overall mitochondrial protein abundance (with the notable exception of tail-anchored proteins) is increased relative to the non-mitochondrial proteome upon knockdown of SAE1. If mitochondrial protein content is increased, then there will be more of the signal-anchored transcription factor and more of the signal-anchored protease, and thus increased TF release and reporter production. More focused validation experiments will be needed to confirm this.

Reviewer #3 (Recommendations for the authors):It would be good to have the study quickly published to guide people in the community who are going to develop screening approaches for their interested directions.Two suggestions:1. In Figure 2A, the authors proposed three possibilities of the effect of the sgRNA: 1. blocking TA targeting of the protease, 2. affecting the level of the protease, and 3. no effect. I would suggest adding the fourth possibility. It is also likely that the sgRNA may also affect the correct targeting of the membrane anchored TF. To my understanding, as long as the protease is not able to meet the TF, no mCherry could be produced. Again this possiblity could be controlled by the SA experiments the authors have performed.

Yes, we agree and have edited the manuscript to describe that this is another possible way by which the HiLITR signal could be reduced. As you mention, the purpose of using multiple screening configurations, such as the SA screen in addition to the TA screen, was to eliminate nonspecific hits as well as false positives that interfere with the membrane anchored TF (which would cause HiLITR signal reduction in both the TA and SA screens and not be of interest).

2. In Figure S4G, could the authors explain why the mito protease generated more mCherry than the peroxisome protease，since the TF is located on the peroxisome?

We believe this result is a consequence of two factors. First, despite several attempts, we were unable to generate a peroxisomal protease that localizes exclusively to peroxisomes. Imaging of the protease in Figure 1 —figure supplement 4F suggests it is also localized to some extent to the mitochondria and/or ER, although this was not directly confirmed. Second, HiLITR activation increases with greater protease expression, and the mitochondrial protease is expressed at higher levels than the peroxisomal protease (compare 1st and 2nd FACS plots in row 2 of Figure 1 —figure supplement 4G). We have added this comment to the figure supplement.